# Development of time-varying global gridded Ts-Tm model for precise GPS-PWV retrieval

Peng Jiang[1, 2], Shirong Ye[2], Yinhao Lu[1], Yanyan Liu[3, 2], Dezhong Chen[2], Yanlan Wu[1]

[1]School of Resources and Environmental Engineering, Anhui University, Hefei, Anhui, China,

[2]GNSS Research Center, Wuhan University, Wuhan, Hubei, China,

[3]Shenzhen Key Laboratory of Spatial Smart Sensing and Services, College of Civil Engineering, Shenzhen University, Shenzhen, Guangdong, China

*Correspondence to:* Peng Jiang (jiangpeng@ahu.edu.cn)

**Abstract:** Water-vapor-weighted mean temperature, $T_m$, is the key variable for estimating the mapping factor between GPS zenith wet delay (ZWD) and precipitable water vapor (PWV). For the near real-time GPS-PWV retrieving, estimating $T_m$ from surface air temperature $T_s$ is a widely used method because of its high temporal resolution and a fair degree of accuracy. Based on the estimations of $T_m$ and $T_s$ at each reanalysis grid node of the ERA-Interim data, we analyzed the relationship between $T_m$ and $T_s$ without data smoothing. The analyses demonstrate that the $T_s$–$T_m$ relationship has significant spatial and temporal variations. Static and time-varying global gridded $T_s$–$T_m$ models were established and evaluated by comparisons with the radiosonde data at 723 radiosonde stations in the Integrated Global Radiosonde Archive (IGRA). Results show that our global gridded $T_s$–$T_m$ equations have prominent advantages over the other globally applied models. At over 17% of the stations, errors larger than 5 K exist in the Bevis equation (Bevis et al., 1992) and in the latitude-related linear model (Yao et al., 2014b), while these large errors are removed in our time-varying $T_s$–$T_m$ models. Multiple statistical tests at the 5 % significance level show that the time-varying global gridded model is superior to the other models at 60.03 % of the radiosonde sites. The second-best model is the 1º × 1º GPT2w model, which is superior at only 12.86 % of the sites. More accurate $T_m$ can reduce the contribution of the uncertainty associated with $T_m$ to the total uncertainty of GPS-PWV, and the reduction augments with the growth of GPS-PWV. Our theoretical analyses with high PWV and small uncertainty in surface pressure indicate that the uncertainty associated with $T_m$ can contribute more than 50 % of the total GPS-PWV uncertainty by using the Bevis equation, and it can decline to less than 25 % by using our time-varying $T_s$–$T_m$ model. However, the uncertainty associated with surface pressure dominates the error budget of PWV (more than 75 %) when the surface pressure has error larger than 5 hPa. GPS-PWV retrievals using different $T_m$ estimates were compared at 74 International GNSS Service (IGS) stations. At 74.32% of the IGS sites, the relative differences of GPS-PWV are within 1 % by applying the static or the time-varying global gridded $T_s$–$T_m$ equations, while the Bevis model, the latitude-related model and the GPT2w model perform the same at respectively 37.84 %, 41.89 % and 29.73 % of the sites. Compared with the radiosonde PWV, the error reduction in the GPS-PWV retrieval by using a more accurate $T_m$ parameterization can be around 1~2 mm, which accounts for around 30 % of the total GPS-PWV error.

## 1. Introduction

Water vapor is an important trace gas and one of the most variable components in the troposphere. The transport, concentration, and phase transition of water vapor are directly involved in the atmospheric radiation and hydrological cycle. It plays a key role in many climate changes and weather processes (Adler et al., 2016; Mahoney et al., 2016; Song et al., 2016).

However, water vapor has high spatial-temporal variability, and its content is often small within the atmosphere. It is a challenge to measure water vapor content accurately and timely. For decades, several methods have been studied, such as radiosondes and water vapor radiometers, sun photometers, and GPS (Campmany et al., 2010; Ciesielski et al., 2010; Liu et al., 2013; Perez-Ramirez et al., 2014; Li et al., 2016). Compared with the traditional water vapor observations, ground-based GPS water vapor measurement has the advantages of high accuracy, high spatial-temporal resolution, all-weather availability, and low-cost (Haase et al., 2003; Pacione and Vespe, 2008; Lee et al., 2010; Means, 2013; Lu et al., 2015). Ground-based GPS water vapor products, mainly including precipitable water vapor (PWV), are widely used in many fields such as real-time vapor monitoring, weather and climate research, and numerical weather prediction (NWP) (Van Baelen and Penide, 2009; Karabatic et al., 2011; Rohm et al., 2014; Adams et al., 2017).

GPS observations require some kinds of meteorological elements to estimate PWV. Zenith hydrostatic delay (ZHD) can be calculated using surface pressure $P_s$ by equation (Ning et al., 2013) :

$$ZHD = \left(2.2767 \pm 0.0015\right)\frac{P_s}{f\left(\varphi, H\right)} \tag{1}$$

where $\varphi$ is the latitude, $H$ is the geoid height in meters, and

$$f\left(\lambda, H\right) = \left(1 - 2.66 \times 10^{-3} \boldsymbol{cos}\,\varphi - 2.8 \times 10^{-7} H\right) \tag{2}$$

Then, zenith wet delay (ZWD) is generated by subtracting ZHD from zenith total delay (ZTD). ZTD can be directly estimated from precise GPS data processing. Finally, a conversion factor $Q$, which is used to map ZWD onto PWV, is determined by the water-vapor-weighted mean temperature $T_m$ over a GPS station. The mapping function from ZWD to PWV is expressed as (Bevis et al., 1992):

$$PWV = \frac{ZWD}{Q} = \frac{ZTD - ZHD}{Q} \tag{3}$$

and $Q$ is computed using following formula:

$$Q = 10^{-6} \rho_w R_v \left[\left(k_3 / T_m\right) + k_2'\right] \tag{4}$$

where $\rho_w$ is the density of liquid water, $R_v$ is the specific gas constant for water vapor, $k_2' = (22.1 \pm 2.2)\mathrm{K} \cdot \mathrm{mbar}^{-1}$ and $k_3 = (3.739 \pm 0.012) \times 10^5 \mathrm{K}^2 \cdot \mathrm{mbar}^{-1}$ are physical constants (Ning et al., 2016) . $T_m$ is the weighted mean temperature which

is defined as a function related to the temperature and water vapor pressure. It can be approximated as the following formula (Bevis et al., 1992):

$$T_m = \frac{\int \frac{e}{T} dz}{\int \frac{e}{T^2} dz} \approx \frac{\sum_{i=1}^{n} \frac{e_i}{T_i} \Delta z_i}{\sum_{i=1}^{n} \frac{e_i}{T_i^2} \Delta z_i} \qquad (5)$$

where $e$ and $T$ respectively represent vapor pressure in hPa and temperature in Kelvin, $i$ denotes the $i$th level and $\Delta z_i$ is the height difference of $i$th level . Vapor pressure $e$ is calculated using equation $e=e_s \times RH$; $RH$ is the relative humidity, and the saturation vapor pressure $e_s$ can be estimated from the temperature observations using Goff-Gratch formula (Sheng et al., 2013).

There are three main approaches to estimate $T_m$. They have respective advantages and disadvantages when they are applied for different purposes:

(1) The integration of vertical temperature and humidity profiles are believed to be the most accurate method. The profile data can be extracted from radio soundings or NWP datasets (Wang et al., 2016). However, some inconveniences have to be endured. It usually takes considerable amounts of time to acquire the NWP data, which is normally released with large volumes every 6 hours. This limits the use of NWP data in the near real-time GPS-PWV retrieving. Radiosonde data is another profile data source, but it has low spatial and temporal resolution. At most of the radiosonde sites, sounding balloons are daily cast at 00:00 UTC and 12:00 UTC. Furthermore, a large amount of GPS stations are not located close enough to the radio sounding sites. Therefore, such methods are appropriate for climate research or the study of long-term PWV trends, but do not meet the real-time requirements.

(2) Several global empirical models of $T_m$ are established based on the analyses of $T_m$ time series from NWP datasets or other sources (Yao et al., 2012; Chen et al., 2014; Bohm et al., 2015). $T_m$ at any time and any location can be estimated from these models. They are often independent of the current meteorological observations which are required to be observed together with the GPS data. However, some important real variations, which may be dramatic during some extreme weather events, can be lost without the constraints of current real data (Jiang et al., 2016). Therefore, these modeled estimates are not accurate enough for high-precision meteorological applications, such as providing GPS-PWV estimates for weather prediction.

(3) Many studies indicated that the $T_m$ parameter has a relationship with some surface meteorological elements, such as surface air temperature or surface air humidity (Bevis et al., 1992; Yao et al., 2014a). These surface meteorological parameters can be measured accurately and rapidly. $T_m$ is then estimated using these surface measurements. However, these studies also revealed that the relationships are often weak, except the $T_s$-$T_m$ relationship. For example, Bevis et al. (1992) introduced the equation $T_m=0.72 T_s+70.2$ [K] after analyzing 8712 radiosonde profiles collected at 13 sites in the U.S. over two years. This equation has been widely used in many other studies.

According to Rohm et al. (2014), GPS-ZTD can be estimated very precisely by real-time GPS data processing. This means that $T_m$ is one of the key parameters in the near real-time GPS-PWV estimation. On the other hand, method (3) is the most suitable method for estimating $T_m$ in near real-time because of its balance between timeliness and accuracy. The $T_s$-$T_m$ relationship has spatial-temporal variations. Several regional $T_s$-$T_m$ equations were established using the profile data over corresponding fields (Wang et al., 2012). However, a $T_s$-$T_m$ model without spatial variation is not good enough for a vast field, e.g. the Indian region (Singh et al., 2014). Aside from this, some vast areas have no specific high-precision $T_s$-$T_m$ model, for example over the oceans. In general, significant differences exist between oceanic and terrestrial atmospheric properties, especially near the surface layer and within the boundary layer. The change of $T_s$ from land to ocean may be very different from that of $T_m$. Therefore it is necessary to model the $T_s$-$T_m$ relationship over oceanic regions, since several ocean-based GPS meteorology experiments demonstrated the potential of such technique to retrieve PWV over the broad ocean (Rocken et al., 2005; Kealy et al., 2012). A global gridded $T_s$-$T_m$ model has been established by Lan et al. (2016). In this model, the $2.0° \times 2.5°$ $T_m$ data from "GGOS Atmosphere" and the $0.75° \times 0.75°$ $T_s$ data from the European Centre for Medium-Range Weather Forecasts (ECMWF) reanalysis data are both smoothed to the resolution of $4° \times 5°$. However, the $T_s$-$T_m$ relationship is varying in time (Yao et al., 2014a), while the Lan et al. (2016) model is static.

The objective of this study is mainly to (1) develop global gridded $T_s$-$T_m$ models without any smoothing of the data, then assess their precision, and (2) study the performances of GPS-PWV retrievals using our $T_s$-$T_m$ models. Table 1 lists the main differences between the $T_s$-$T_m$ model developed in this study and the other global used $T_m$ models. In section 2, the data sources and determining methods of $T_m$ are introduced in detail. Then, in section 3 we analyze the $T_s$-$T_m$ relationships and their variations on a global scale. Global-gridded $T_s$-$T_m$ estimating models in different forms are established and evaluated in section 4. Section 5 assesses the accuracies of different PWV retrievals and section 6 presents conclusions based on our experiments.

**Table 1**.  Main differences between $T_s$-$T_m$ models developed in this study and other global used $T_m$ estimation models

| Strategies \ $T_s$-$T_m$ Models | Bevis model (Bevis et al., 1992) | Latitude-related linear model (Yao et al., 2014b) | Global-gridded model (Lan et al., 2016) | Time-varying global gridded model (our study) | GPT2w model (Bohm et al., 2015) |
|---|---|---|---|---|---|
| **Applicable Regions** | Regional/Global | Global | Global | Global | Global |
| **Data Sources** | Radiosonde | $T_s$ from the $0.75° \times 0.75°$ ERA-Interim, and $T_m$ from the $2° \times 2.5°$ "GGOS Atmosphere" | $T_s$ from the $0.75° \times 0.75°$ ERA-Interim, and $T_m$ from the $2° \times 2.5°$ "GGOS Atmosphere" | $T_s$ and $T_m$ both from the $0.75° \times 0.75°$ ERA-Interim | $T_m$ from the $1° \times 1°$ ERA-Interim monthly mean data |
| **Data Processing** | Integrate radiosonde profiles | $4° \times 5°$ Sliding window smooth | $4° \times 5°$ Sliding window smooth | Integrate ERA-Interim profiles | Integrate ERA-Interim profiles |
| **Variations in model** | Static without any variations | Spatial variations depend on only latitude(15° latitude interval), but no temporal variations | $4° \times 5°$ global gridded, but no temporal variations | $0.75° \times 0.75°$ global gridded and considering time variations | $1° \times 1°$ global gridded, considering time variations, but independent of |

| | | | | | current surface |
|---|---|---|---|---|---|
| | | | | | observations |

## 2. Data Sources and Methodology

As the definition of $T_m$ in equation (5), $e_i$ parameter at the middle height of $i$th level is calculated by vertically exponential interpolation of the water vapor pressure of its two neighbor measurement points. The temperature is estimated by linear interpolation of the two neighbor temperatures. The integral intervals are from the earth surface to the top level of profile data. The height of top level depends on the data sources we employed. The essential profile data, including the temperature, height and relative humidity values through the entire atmospheric column, can be obtained from the radiosondes or NWP datasets.

We employed radiosonde data from the Integrated Global Radiosonde Archive (IGRA, ftp://ftp.ncdc.noaa.gov/pub/data/igra) to calculate $T_m$. Version 2.0 of the IGRA-derived sounding parameters provides pressure, geopotential height, temperature, saturation vapor pressure, and relative humidity observations at the observed levels. Bias may be introduced if the integrals were terminated at lower levels (Wang et al., 2005), thus the integrations were performed up to the topmost valid radiosonde data. According to our quality control processes, some radiosonde profile data were rejected. In each profile, the surface observations must be available and the top profile level should not be lower than 300 hPa standard level. Furthermore, the level number between the surface and the top level should be greater than 10 to avoid too sparse vertical profiles. At most of the radio sounding stations, sounding balloons are launched every 12 hours, and their ascending paths are assumed to be vertical.

Profile data are usually provided by NWP products at certain vertical levels. The ERA-Interim product from ECMWF provides data on a regular 512 longitude by 256 latitude N128 Gaussian grid after the grid transforming performed by the NCAR Data Support Section (DSS). On each grid node of ERA-Interim, temperature, relative humidity and geopotential at 37 isobaric levels from 1000 hPa to 1 hPa can be obtained. Dividing the geopotential by constant gravitational acceleration value ($g \approx 9.80655$ m/s$^2$), we can determine the geopotential heights of the surface and levels. Datasets are available at 00:00, 06:00, 12:00 and 18:00 UTC every day and have been covering a period from 1979.01 to present.

In theory, the computation of equation (5) should be integrated through the entire atmospheric column, and the geopotential height should be converted to the geometric height. However, water vapor is solely concentrated in the troposphere, and most of it is specifically located within the first 3 kilometers above sea-level. Moreover, in the two selected datasets, the geopotential heights of top pressure levels are approximately 30~40 km. Geopotential height is very close to geometric height in such height ranges. According to our computation, the relative difference between them is only between 0.1 %~0.9 %. In fact, the height difference $\Delta z$ can be replaced by the geopotential height difference $\Delta h$ in equation (5), since the division can almost eliminate the difference between the two different height types. The $T_m$ value nearly has no change after such height replacement. For the convenience of calculation, we directly employed the geopotential height

variable. In this paper, we denoted the $T_m$ derived from ERA-Interim as $T_{m\_ERAI}$.

At each reanalysis grid node, the computation of equation (5) always starts from the surface height to the top pressure level. The pressure levels below surface height were rejected. $T_s$ is defined as the variable of "temperature at 2 meters above ground", and surface water vapor pressure can be derived from the "2 meter dewpoint temperature" variable in ERA-Interim. These $T_s$ were also used in the regression analyses between $T_s$ and $T_m$.

## 3. Correlation between Ts and Tm

Many studies have indicated the close relationship between $T_s$ and $T_m$. However, $T_m$ is also found not being closely related to $T_s$ in some regions, e.g., in the Indian zone (Raju et al., 2007). Using the $T_m$ and $T_s$ generated from the global gridded reanalysis data, we are able to study the $T_s$-$T_m$ relationship in detail.

We first carried out a linear regression analysis on four years of $T_s$ and $T_m$ data generated from the radiosonde data and the global gridded ERA-Interim datasets, with data covering the period 2009\01 to 2012\12. The analysis results are shown in figure 1. Although the two datasets have different temporal resolutions (12 hours for the radiosonde data and 6 hours for the ERA-Interim data) and spatial resolutions, both analyses agree well with each other. This is expected because the radiosonde data have been assimilated into the ERA-Interim products. Our analyses also indicate that the $T_s$-$T_m$ correlation coefficient is generally related to the latitude. The same conclusion has been drawn in other studies (Yao et al., 2014b). Significant positive correlation coefficients can be found at mid- and high- latitudes and reach a maximum in the polar regions. The correlation coefficients drop dramatically at low latitudes. This is because $T_m$ is stable there, showing independency of the other parameters. To study the variations of $T_s$ and $T_m$, we illustrated the denary logarithm values of their standard deviations in figure 2. It is evident that $T_m$ varies to a lesser degree than $T_s$ at low latitudes. Aside from the latitude-related features, there are obvious differences of the $T_s$-$T_m$ correlation coefficients between land and ocean. We even found that negative correlation coefficients over certain oceans, e.g., low-latitude Western Pacific, Bay of Bengal or Arabian Sea (see figure 1). Unreliable regression analysis results may be derived when the $T_s$ and $T_m$ data both have small variations. In figure 3, scatter plots of $T_s$ and $T_m$ from ERA-Interim at two locations 0.35° N 180.00° E and 70.53° N 180.00° E are given. As the blue dots show, the $T_s$-$T_m$ relationship is weak in the areas near the equator, because the entire variation ranges of $T_s$ and $T_m$ are both within 10 K. This results in a meaningless linear regression (see the magenta line). The $T_s$-$T_m$ correlation coefficient is only -0.0893 there. Other than the large spatial variations, studies have revealed that the $T_s$-$T_m$ relationship also has temporal variations (Wang et al., 2005). Therefore, a good $T_s$-$T_m$ model should take both the spatial and temporal variations into consideration, and this is the main aim in the following sections.

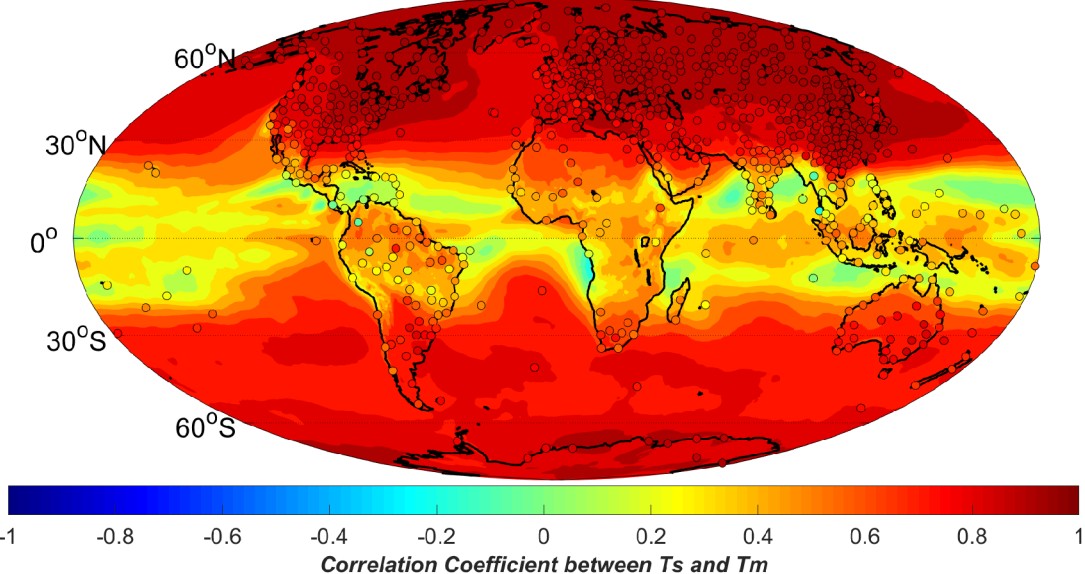

**Figure 1: Correlation coefficients between $T_s$ and $T_m$ generated from radiosonde data (dots) and ERA-Interim reanalysis datasets (color-filled contours) over a period of 4 years from 2009 to 2012.**

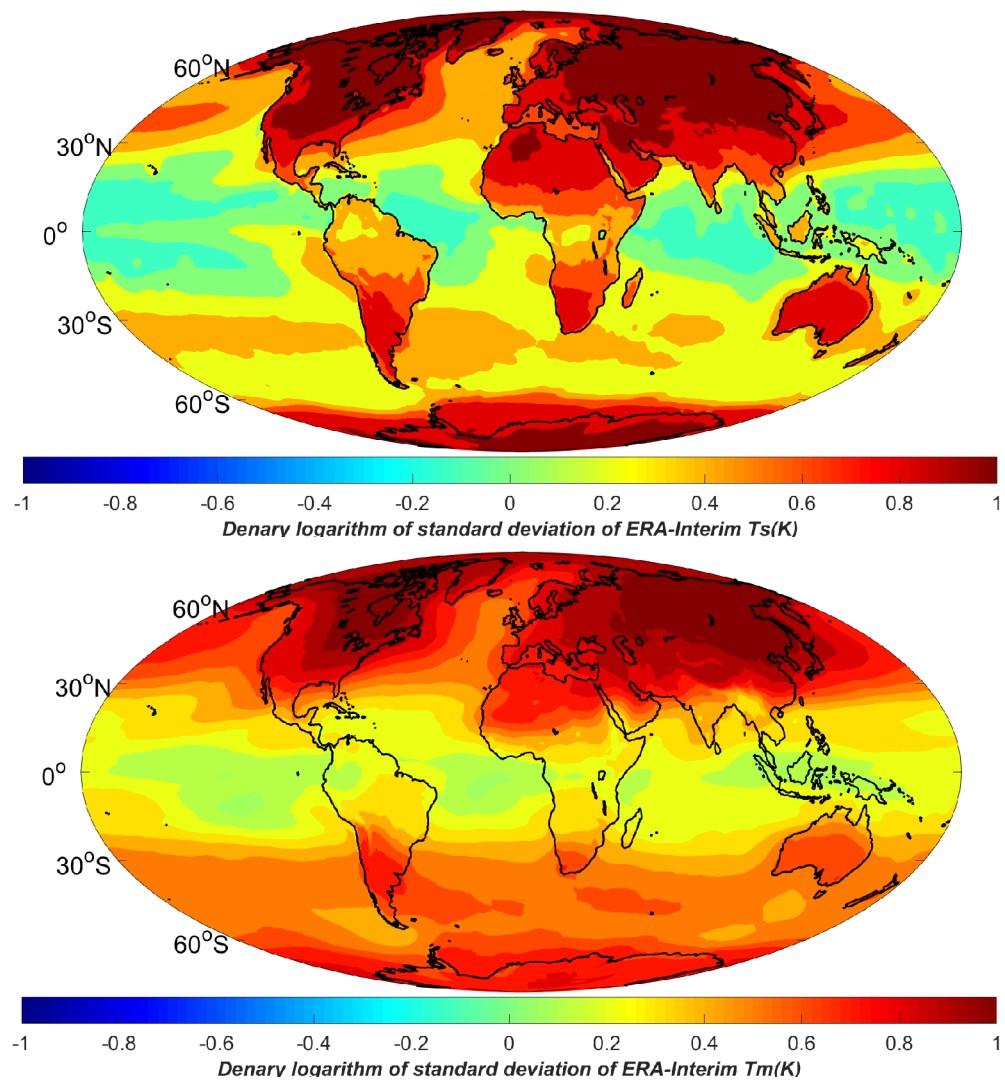

**Figure 2: Denary logarithm of the standard deviation of (top) $T_s$ and (bottom) $T_m$ generated from the ERA-Interim data covering the year 2009 to 2012. Temperature unit is Kelvin.**

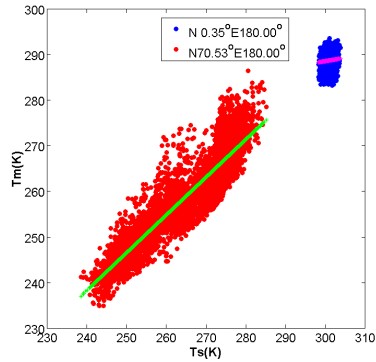

**Figure 3:** *$T_s$-$T_m$* **scatter plots at two locations: (blue dots) 0.35° N 180.00° E and (red dots) 70.53° N 180.00° E, the magenta and green lines are their linear fitting curves. Temperature unit is Kelvin.**

## 4. Development of global-gridded *$T_s$-$T_m$* models

Since the *$T_s$-$T_m$* relationship has large spatial variations, a global gridded *$T_s$-$T_m$* model is preferred for precise GPS-PWV

estimations. In this section, a static global gridded model and a time-varying global gridded model are established and assessed.

### 4.1 Static global-gridded *$T_s$-$T_m$* model

     A linear formula $T_m = aT_s + b$ for the relation between $T_m$ and $T_s$ has been adopted in many studies. Based on the $T_s$ and

$T_m$ products from the ERA-Interim data covering the year 2009 to 2012, we performed linear fittings of $T_m$ versus $T_s$ on each

grid point. Then, the slope constant (*a*), the intercept constant (*b*) and the fitting root mean square error (RMSE) of each linear

expression were calculated and contoured in figure 4. The *a* and *b* values are related to the latitude as well as the underlying

surface (e.g. land, ocean). In the mid-high latitudes over the Northern Hemisphere, constant *a* value varies from 0.6 to 0.8, and

constant *b* is approximately 100~50 over most of the continents. The constants in the Bevis equation are within these value

ranges. Constant *a* is smaller (approximately 0.5~0.7) over land at the mid to high latitudes over the Southern Hemisphere.

Especially, there are abrupt changes in the values of constants *a* and *b* from land to ocean at the mid to high latitudes due to

the different variation features of $T_s$ and $T_m$ (see figure 2). At the low latitudes, the *a* value is smaller than over the other regions,

because of the low variations of $T_s$ and $T_m$. The fitting RMSEs are within 2~4 K over the mid to high latitude lands, and lower

values are obtained over the oceans or at the low latitudes. The reason for the low RMSE around the equator is the smaller

fluctuation of $T_m$. Meanwhile, there is no RMSE larger than 4.5 K in the results of our model. As we did not perform any

spatial or temporal smoothing of the data during the data processing, both the precision and resolution of our static model is

better than other models (e.g. Lan et al., 2016).

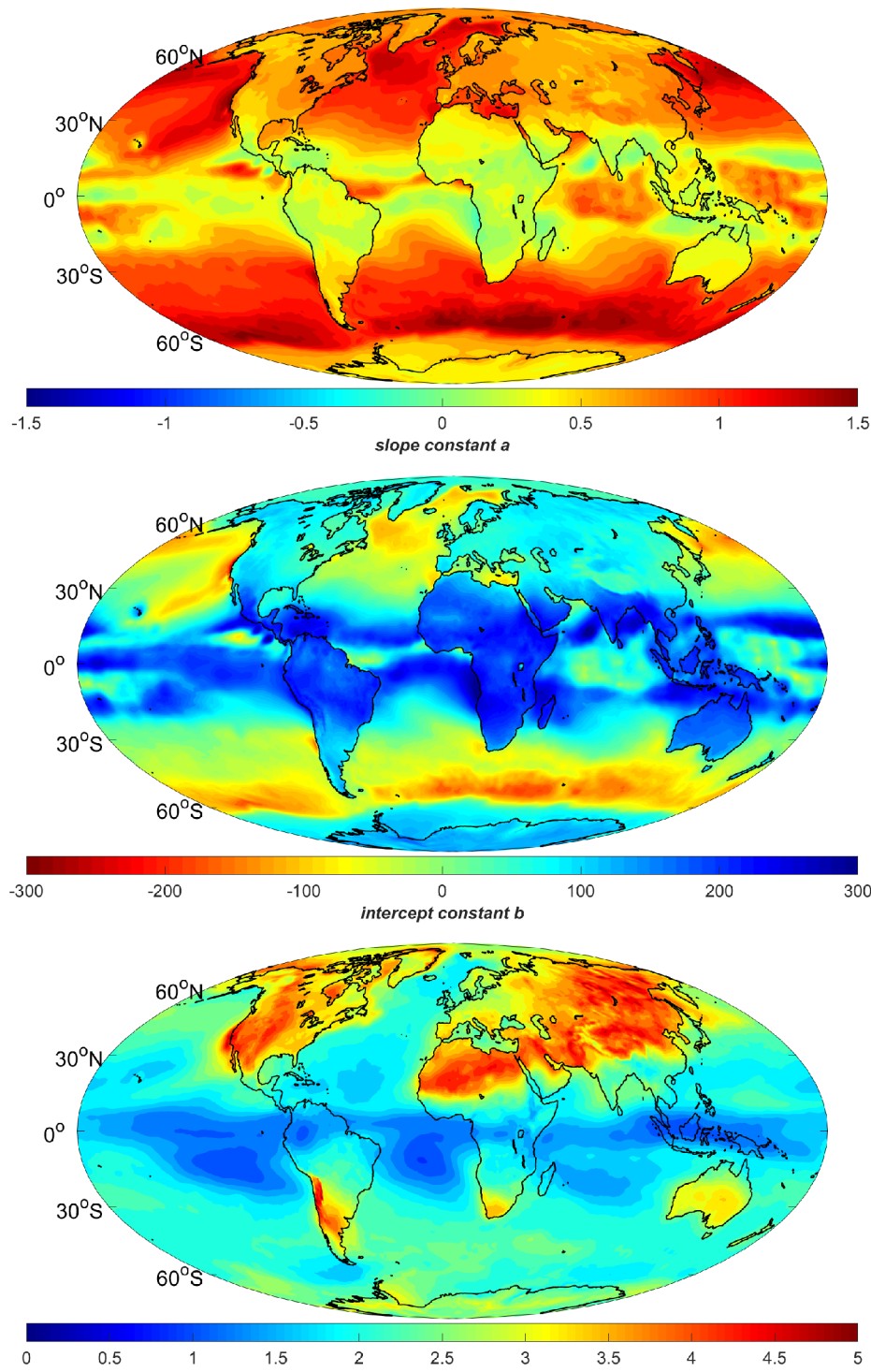

**Figure 4: Distributions of the (top) slope constant *a*, (middle) intercept constant *b*, and (bottom) RMSE of static linear $T_s$-$T_m$ equations at ERA-Interim grid nodes. Temperature unit is Kelvin.**

## 4.2 Time-varying global-gridded $T_s$-$T_m$ model

The time variation of $T_s$-$T_m$ relationship should also be considered in a precise $T_s$-$T_m$ model. Therefore, a time-varying equation is applied for $T_s$-$T_m$ regression at each grid node:

$$T_m = aT_s + b + m_1\, cos\left(\frac{doy}{365.25}2\pi\right) + m_2\, sin\left(\frac{doy}{365.25}2\pi\right) + n_1\, cos\left(\frac{doy}{365.25}4\pi\right) +$$

$$n_2\, sin\left(\frac{doy}{365.25}4\pi\right) + p_1\, cos\left(\frac{hr}{12}\pi\right) + p_2\, sin\left(\frac{hr}{12}\pi\right) \tag{6}$$

where *doy* represents the observed day of year and *hr* is the observed hour in UTC time; $(m_1, m_2)$, $(n_1, n_2)$ and $(p_1, p_2)$ are fitting coefficients. These equations can reflect the amplitudes of annual, semiannual and diurnal variations in our $T_s$-$T_m$ models.

Our new regression model found similar values for the coefficients *a* and *b* (of its static term) as for the static model in section 4.1, except for some differences over the oceans. In figure 5, besides these constants *a* and *b*, we also illustrate the amplitudes of annual, semiannual, and diurnal terms. We can see that there are large annual variations (amplitude > 5 K) in the vast regions from Tibet to North Africa, and in some places of the Siberia and Chile. Large diurnal variations (amplitude > 3 K) mainly occur over the mid-latitude lands such as Northeast Asia or North America. Semiannual variations, however, are small in most of areas except some high-latitudes (amplitude > 3 K). All variations are smaller over the oceans due to the slower temperature changes over water than over land. The estimated $T_m$ RMSE is also contoured in figure 5, and we can see that the RMSE dropped significantly in the regions with large annual or diurnal variations.

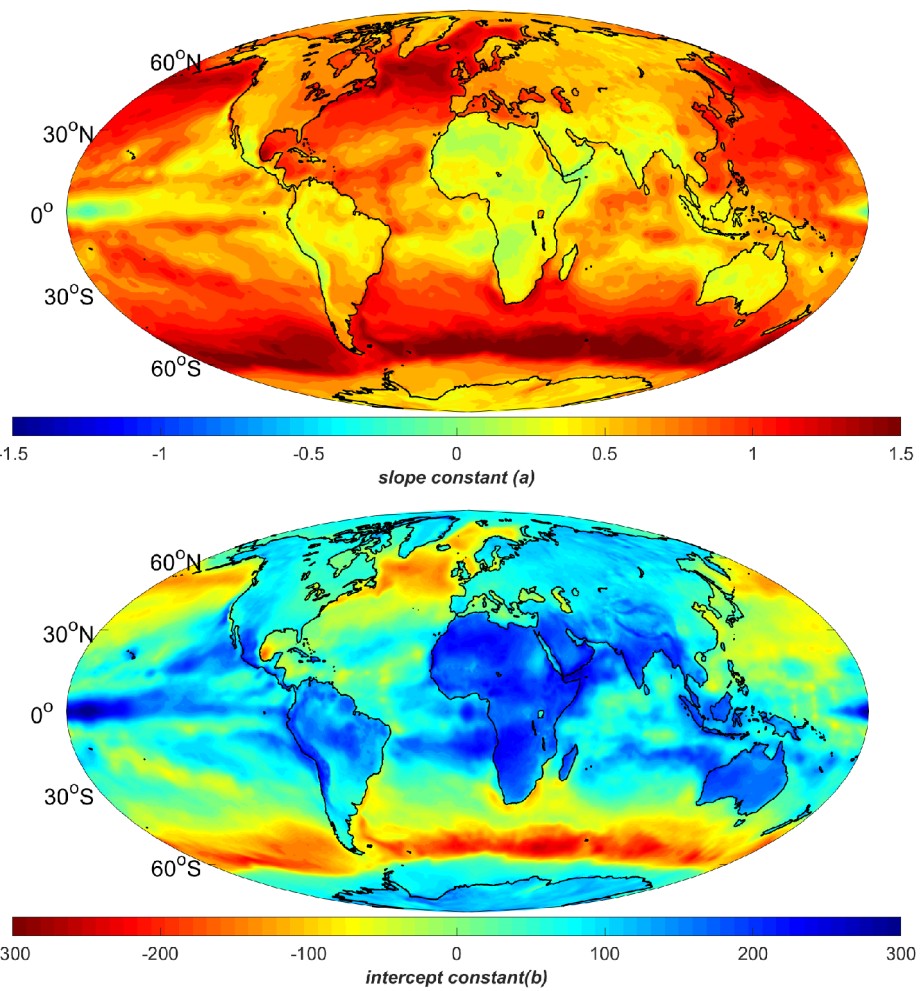

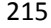

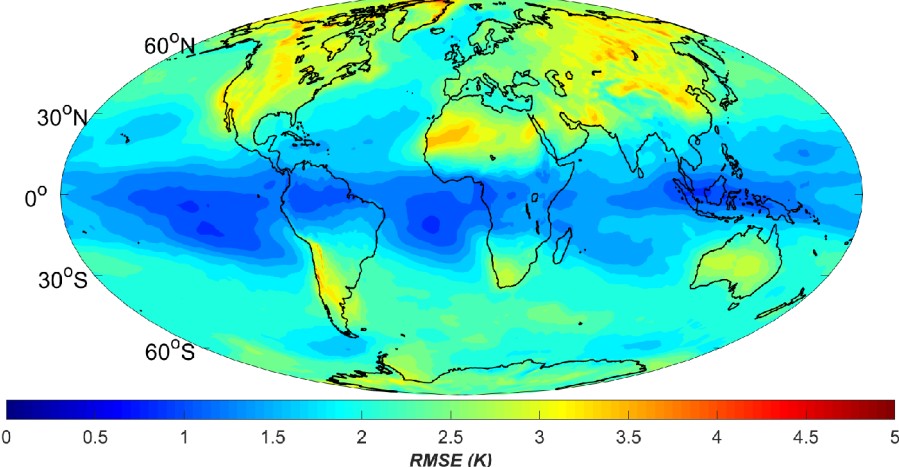

**Figure 5:** (top)**The slope constant** *a*, **(second) intercept constant** *b*, **amplitudes of** $T_m$ **(third) annual, (forth) semiannual and (fifth)diurnal terms in our time-varying global gridded** $T_s$ **-**$T_m$ **model, and (bottom) the model estimated** $T_m$ **RMSE distribution. Temperature unit is Kelvin.**

### 4.3 Assessments of $T_s$-$T_m$ models

To further assess the precision of the $T_s$-$T_m$ models using other independent data sources, we generated $T_m$ and $T_s$ from the radiosonde data at 723 radiosonde stations in the year 2016. These data are not assimilated into the 2009~2012 ERA-Interim datasets. As a result, we can regard them as independent of our model. At each radiosonde site, different $T_s$-$T_m$ models were employed to calculate $T_m$. In addition, we also estimated $T_m$ using the 1° × 1° GPT2w model (Bohm et al., 2015), which is a global gridded $T_m$ empirical model independent of the surface meteorological observation data. Then, these calculated $T_m$ will be evaluated by comparing them with the integrated $T_m$ of radiosondes (denoted as $T_{m\_RS}$) twice a day.

The model estimations of $T_m$ are denoted as $T_{m\_Bevis}$, $T_{m\_LatR}$, $T_{m\_static}$, $T_{m\_varying}$, and $T_{m\_GPT2w}$ from respectively the Bevis equation, the latitude-related model, our static global gridded model, time-varying global gridded model, and the GPT2w model. When the global gridded models are employed, the radiosonde station may not be located at a grid node. Therefore, we interpolated the coefficients in the $T_s$-$T_m$ equations from the neighboring grids to the radiosonde sites. The interpolation formula is expressed as (Jade and Vijayan, 2008) :

$$C_{site} = \sum_{i=1}^{4} w^i C_{grid}^i \tag{7}$$

$C_{site}$ and $C_{site}^i$ represent the coefficients in $T_s$-$T_m$ equations at the radiosonde site location and its neighboring grids, respectively. $w^i$ are the interpolation coefficients, which are determined using the equation:

$$w^i = \frac{\left(R\psi^i\right)^{-\lambda}}{\sum_{j=1}^{4}\left(R\psi^j\right)^{-\lambda}} \tag{8}$$

where $R$=6378.17 km is the mean radius of the earth, $\lambda$ is the scale factor which equals one in our study, and $\psi^j$ is the

angular distance between the *i*th grid node and the station's position. $\psi^i$ are computed using following formula ( with latitude $\varphi$ and longitude $\theta$ ):

$$\cos\psi^i = \sin\varphi^i \sin\varphi + \cos\varphi^i \cos(\theta^i - \theta)\cos\varphi \qquad (9)$$

Considering the fact that the reanalysis grids are definite, and every radiosonde site is in situ; we can compute the interpolation coefficients in equation (7) for all of the radiosonde stations. Then, these coefficients are stored as constants to avoid reduplicating the calculation.

Taking $T_{m\_RS}$ as the reference values, we calculated the biases and RMSEs of $T_{m\_Bevis}$, $T_{m\_LatR}$, $T_{m\_static}$, $T_{m\_varying}$ , and $T_{m\_GPT2w}$ at each radiosonde site. The results are illustrated in figure 6. Obviously, in many regions, the Bevis equation has a bad precision with the absolute bias and RMSE both larger than 5 K. $T_{m\_LatR}$ can reduce the estimated biases in many areas, but the RMSEs remain large. Large biases still exist at quite a few radiosonde stations, e.g. in Africa or West Asia. $T_{m\_static}$ and $T_{m\_GPT2w}$ remove the large $T_m$ biases at most of the radiosonde stations. $T_{m\_varying}$ performs significantly better over the world, especially in the Middle East, North America , Siberia region, etc.

Detailed statistics of the distributions of the bias and RMSE using different models are shown in figure 7 and table 2. At over 97.37 % of the radiosonde stations, the biases of $T_{m\_varying}$ are within -3~3 K. Large positive biases (> 3 K) nearly disappear in $T_{m\_varying}$. In contrast, there are significant large biases in $T_{m\_Bevis}$ and $T_{m\_LatR}$. Improvements in RMSE are more evident. The RMSEs of $T_{m\_varying}$ are smaller than 4 K at over 91 % of the radiosonde sites, while few sites (<1 %) have RMSEs larger than 5 K. This is clearly better than the other models. In $T_{m\_Bevis}$ and $T_{m\_LatR}$, there are more than 17 % of the radiosonde sites have RMSEs larger than 5 K. The overall performance of $T_{m\_GPT2w}$ is very close to $T_{m\_Bevis}$, except that its absolute bias is smaller than the other $T_s$-$T_m$ models.

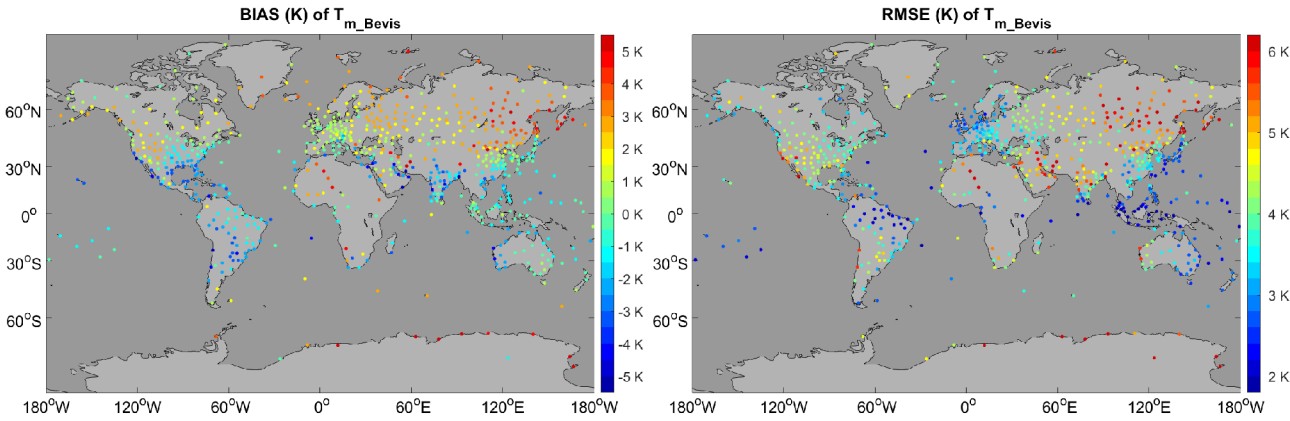

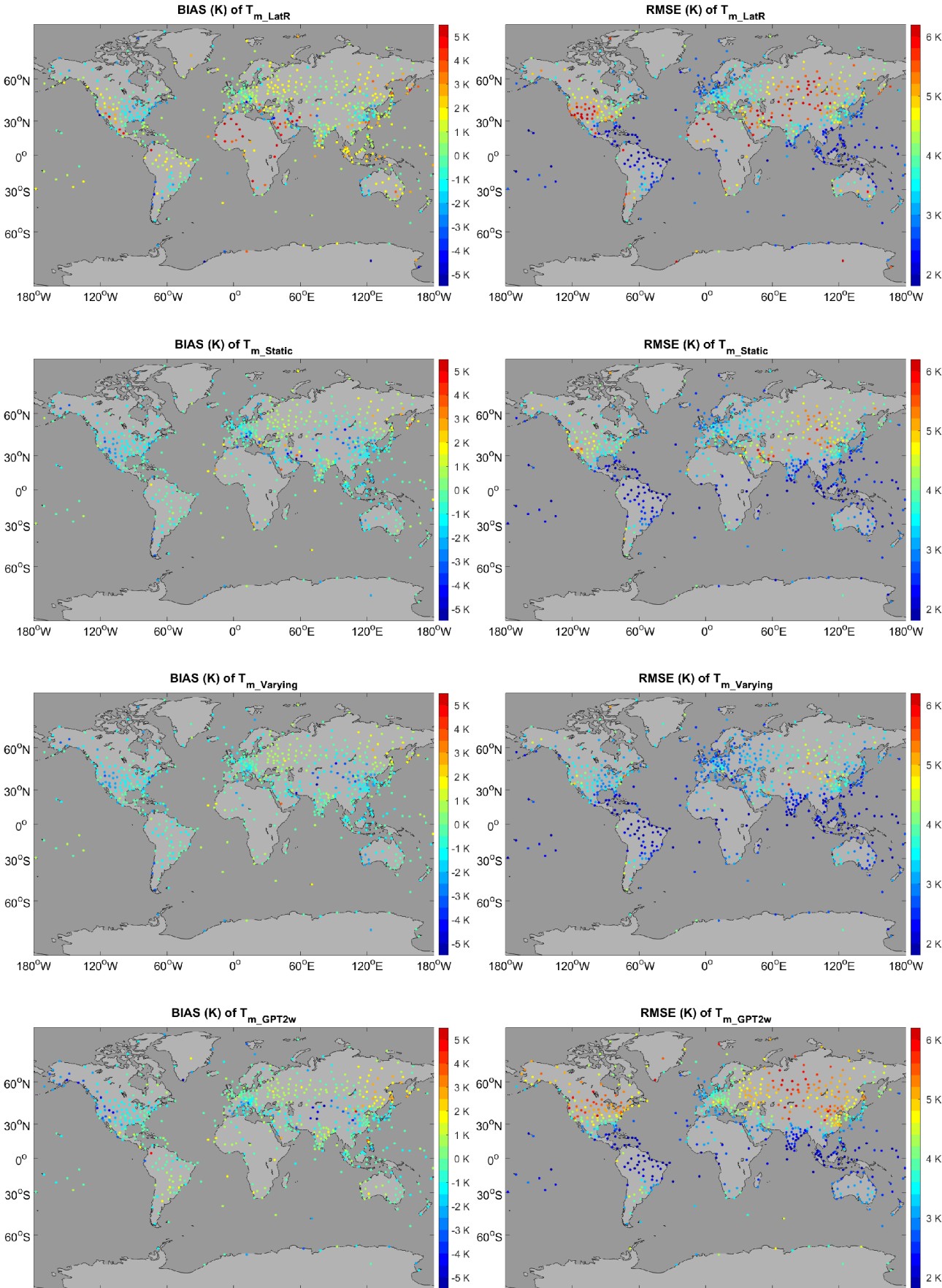

**Figure 6: (left) The bias and (right) the RMSE of the estimated $T_m$ from respectively (top) the Bevis equation, (second) the latitude-related model, (third) our static global gridded model, (forth) our time-varying global gridded model and (bottom) the GPT2w model at each radiosonde station. Reference data are the radiosonde data of the year 2016. Temperature unit is Kelvin.**

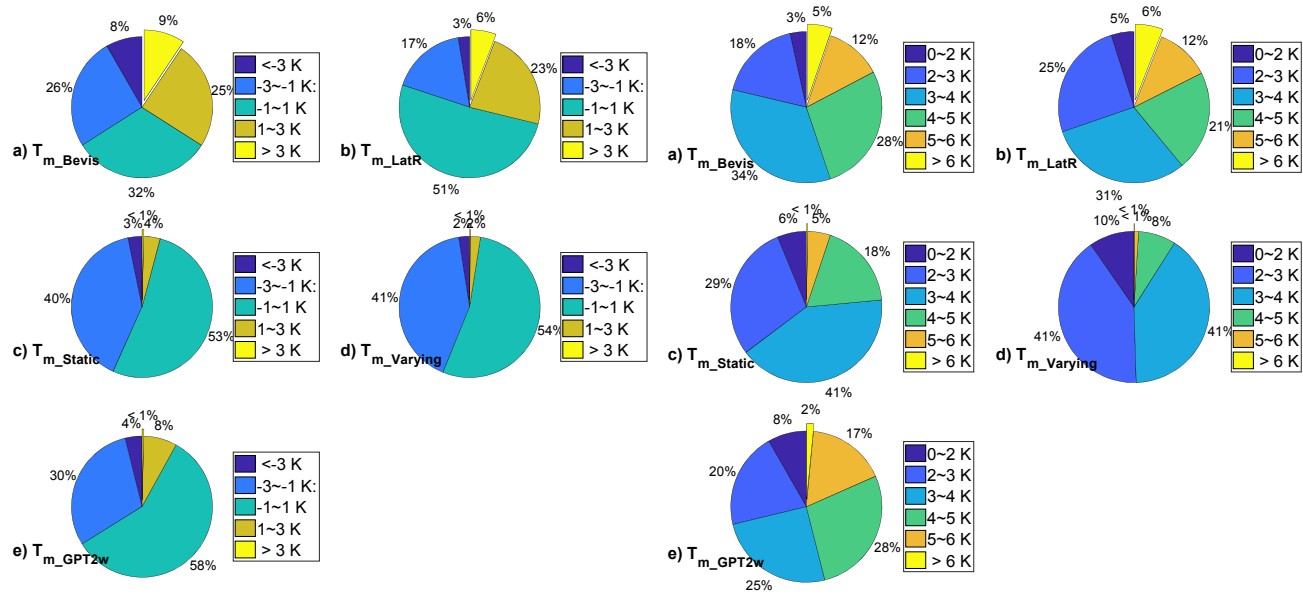

1) Bias distribution                                    2) RMSE distribution

**Figure 7: The distributions of (left) the biases and (right) the RMSEs of $T_{m\_Bevis}$, $T_{m\_LatR}$, $T_{m\_static}$, $T_{m\_varying}$ and $T_{m\_GPT2w}$ compared with the radiosonde data at 723 stations in the year 2016. Temperature unit is Kelvin.**

**Table 2: Statistics of $T_m$ estimates from different models. Reference data are the radiosonde $T_m$ derivations.**

| Statistics | $T_{m\_Bevis}$ | $T_{m\_LatR}$ | $T_{m\_static}$ | $T_{m\_varying}$ | $T_{m\_GPT2w}$ |
|---|---|---|---|---|---|
| Average value of absolute $T_m$ bias (K) | 1.88 | 1.30 | 1.13 | 1.08 | 1.06 |
| Average value of $T_m$ RMSE (K) | 3.95 | 3.81 | 3.36 | 3.01 | 3.80 |
| Average relative RMSE of $T_m$ ( %) | 1.44 | 1.39 | 1.22 | 1.09 | 1.39 |
| Max Relative RMSE of mean $T_m$ ( %) | 3.69 | 4.26 | 2.40 | 2.19 | 4.31 |
| % of sites with $T_m$ RMSE < 4 K | 55.19 | 61.00 | 76.49 | 91.01 | 53.94 |
| % of sites with $T_m$ Relative RMSE less than 1.5 % | 59.47 | 64.73 | 78.01 | 89.76 | 56.43 |

To identify the superior $T_m$ estimation model at each radiosonde site, we employed the following statistical tests under the assumption of a normal distribution of the estimated $T_m$ error:

     (1) First, Brown-Forsythe tests (Brown and Forsythe, 1974) of equality of variances were carried out at each site for estimating the $T_m$ errors from two different models, e.g., model **A** and **B**. The purpose of this step is to determine whether there is significant variance difference between the $T_m$ results. If the test rejects the null hypothesis at a 5 % significance level that

the errors of model **A** and **B** have the same variance, the model with the smaller sample variance is regarded as the better one. However, if the test does not reject the homogeneity of variances, analysis of variance (ANOVA) is performed in the next step.

     (2) ANOVA is a technique used to analyze the differences among group means (Hogg, 1987). It evaluates the null hypothesis that the samples all have the same mean against the alternative that the means are not the same. If the null hypothesis is rejected at a 5 % significance level, the $T_m$ sample with smaller absolute mean value is believed to be better. Otherwise, we

think that two models perform almost the same at this radiosonde site.

     (3) After multiple tests and comparisons, the best model at each radiosonde station may be identified. However, at some sites no superior model can be confirmed. All the models are believed to have equivalent performances.

Finally, we counted the number of sites at which each $T_m$ model respectively performed the best. The results are given in table 3. The time-varying global gridded model is superior to the others at 434 radiosonde stations (60.03 % of all sites), while the second-best estimation, $T_{m\_GPT2w}$, is superior at only 12.86 % of the sites.

**Table 3: Number of radiosonde sites at which the five global applied $T_m$ estimation models respectively perform superiorly**

| Superior model | None | $T_{m\_Bevis}$ | $T_{m\_LatR}$ | $T_{m\_static}$ | $T_{m\_varying}$ | $T_{m\_GPT2w}$ |
|---|---|---|---|---|---|---|
| Number of sites | 50 | 46 | 61 | 39 | 434 | 93 |

In figure 8,the $T_m$ series at the IGRA station No.62378 (29.86° N 31.34° E, in Egypt) are given. We can see that large negative biases (< -5 K) between $T_{m\_Bevis}$ (or $T_{m\_LatR}$) and $T_{m\_RS}$ exist. $T_{m\_static}$ performs only slightly better from July to October. However, $T_{m\_varying}$ and $T_{m\_GPT2w}$ can eliminate most of the seasonal errors. Different properties of $T_m$ series appear at another IGRA station No.40841 (30.25° N 56.97° E, in Iran). Some observation data are missing, but we can still see that there are large positive differences (> 5 K) between $T_{m\_Bevis}$ (or $T_{m\_LatR}$) and $T_{m\_RS}$ throughout the year. The biases of $T_{m\_static}$ are much smaller, but some large errors still appear in many months. The $T_{m\_varying}$, however, performs as well as the $T_m$ calculated from the radiosonde data, with small biases and capturing the variations well. The time series of $T_{m\_GPT2w}$ are smoother and cannot capture the fluctuations of the $T_m$ time series, causing a worse accuracy than $T_{m\_varying}$.

On the other hand, even $T_{m\_varying}$ have large differences from $T_{m\_RS}$ at a few IGRA stations. This can be explained by the fact that our fitting analyses are based on the $T_m$ values derived from ERA-Interim profiles. The quality of ERA-Interim data can be very poor in the regions with sparse observation data (Itterly et al., 2018).

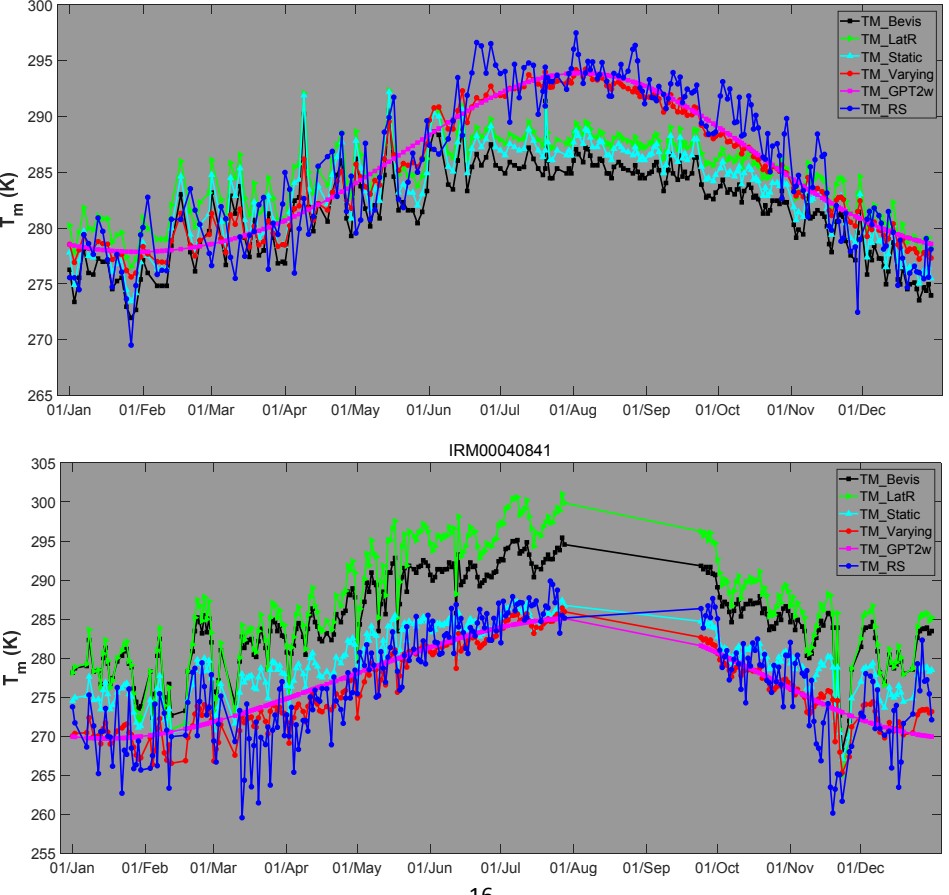

**Figure 8:** $T_m$ **series of** $T_{m\_Bevis}$, $T_{m\_LatR}$, $T_{m\_static}$, $T_{m\_varying}$, $T_{m\_GPT2w}$ **and** $T_{m\_RS}$ **at the IGRA station (top) No.62378 and (bottom) No.40841. Temperature unit is Kelvin.**

## 5. GPS-PWV retrieving experiments

GPS-PWV has different error sources with different properties. It is complicated to evaluate the GPS-PWV uncertainty here due to the lack of collaborated additional independent techniques to monitor water vapor at the GPS site.

### 5.1 Theoretical analysis of the GPS-PWV uncertainty

A comprehensive research on the uncertainty of GPS-PWV has been carried out by Ning et al. (2016). The uncertainties of the ZTD, ZHD and conversion factor $Q$ have been studied in detail. The total uncertainty of GPS-PWV is:

$$\sigma_{PWV} = \frac{1}{Q}\sqrt{\sigma_{ZTD}^2 + \left(\frac{2.2767\sigma_{Ps}}{f(\varphi,H)}\right)^2 + \left(\frac{P_s\sigma_c}{f(\varphi,H)}\right)^2 + \left(PWV \cdot \sigma_Q\right)^2} \qquad (10)$$

where $\sigma_{PWV}$, $\sigma_{ZTD}$, $\sigma_{Ps}$, and $\sigma_Q$ are respectively the uncertainties of GPS-PWV, ZTD estimation, $P_s$ observations and

conversion factor $Q$. $\sigma_c = 0.0015$ denotes the uncertainty of constant $C = 2.2767$ in equation (1), $PWV$ is the value of GPS-PWV, and

$$\sigma_Q = 10^{-6}\rho_w R_v \sqrt{\left(\frac{\sigma_{k_3}}{T_m}\right)^2 + \sigma_{k_2'}^2 + \left(k_3\frac{\sigma_{T_m}}{T_m^2}\right)^2} \qquad (11)$$

where $\sigma_{k_3} = 0.012\times10^5$ K$^2$ hPa$^{-1}$, $\sigma_{k_2'} = 2.2$ K hPa$^{-1}$, and $\sigma_{T_m}$ denote respectively the uncertainties of $k_3$, $k_2'$ and $T_m$ in equation (4). The variation of $\sigma_Q$ with the value of $T_m$ and $\sigma_{T_m}$ is depicted in figure 9. Assuming the $T_m$ is 280 K, we find

that the $\sigma_Q$ increases by over 60 % (from 0.069 to 0.112) as the $\sigma_{T_m}$ raises from 3.0 K to 5.0 K. However, the $\sigma_Q$ is less sensitive to the value of $T_m$. The $\sigma_Q$ raises only by 17.96 % (about from 0.061 to 0.075) as the value of $T_m$ drops from 300 K to 270 K with $\sigma_{T_m} = 3.0$ K.

Ning et al. (2016) assumed the $T_m$ were obtained from NWP models so the uncertainty of $T_m$ was set to be small ($\sigma_{T_m} = 1.1$ K). However, as shown in section 4.3, the uncertainties of $T_m$ from different $T_m$ models are significantly larger at

the radiosonde stations. For each radiosonde station, we calculated the mean value of $T_m$ and assigned the $\sigma_{T_m}$ with the RMSEs of $T_m$ given in figure 6. Then we obtained the $\sigma_Q$ in equation (11). Our statistics indicate that the $\sigma_Q$ using our

varying $T_s$-$T_m$ model decreases by average 19.26 %, 17.77 %, 7.79 % and 18.67 % with respect to the $\sigma_Q$ respectively using the $T_{m\_Bevis}$, $T_{m\_LatR}$, $T_{m\_static}$, and $T_{m\_GPT2w}$. For example, at the IGRA station No.42724 (22.88° N 91.25° E, in India), $\sigma_Q$ drops by 53 % from 0.141 of the $T_{m\_Bevis}$ to 0.066 of the $T_{m\_varying}$.

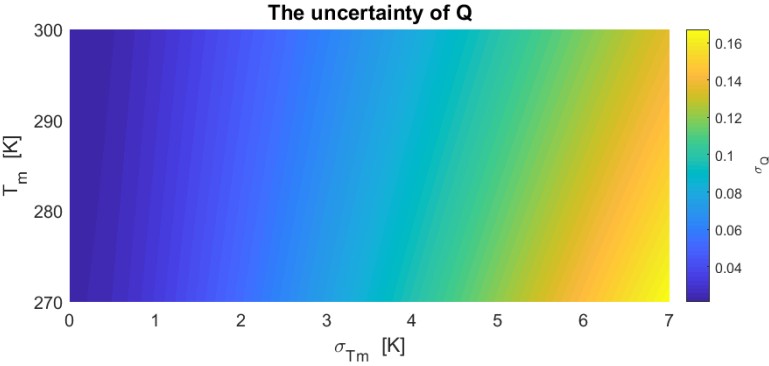

Figure 9. Variation of the uncertainty of Q with the value of $T_m$ and the uncertainty of $T_m$

The uncertainty of $Q$ will be propagated to the total uncertainty of GPS-PWV according to equation (10). We obtained the contributions of the different terms in equation (10) to the total GPS-PWV uncertainty. The contribution of one term is measured by the percentage it accounts for the total $\sigma_{PWV}$. The percentages are computed using formulas:

$$p_{ZTD} = \frac{(\sigma_{ZTD}/Q)^2}{\sigma_{PWV}^2}, \quad p_{Ps} = \frac{\left[2.2767\sigma_{Ps}/(f(\varphi,H)Q)\right]^2}{\sigma_{PWV}^2}, \quad p_C = \frac{\left[P_s\sigma_c/(f(\varphi,H)Q)\right]^2}{\sigma_{PWV}^2}, \quad p_Q = \frac{(PWV\cdot\sigma_Q/Q)^2}{\sigma_{PWV}^2} \quad (12)$$

where $p_{ZTD}$, $p_{Ps}$, $p_C$ and $p_Q$ indicate respectively the contribution of the uncertainty associated with $ZTD$, $P_s$, constant $C$ and factor $Q$ to the total $\sigma_{PWV}$. Following the summaries of Ning et al. (2016), we assumed that $\sigma_{ZTD} = 4\,\text{mm}$ and $\sigma_C = 0.0015$. $T_m$ identically equals to 280 K since the $\sigma_Q$ is less sensitive to the value of $T_m$ with respect to the $\sigma_{Tm}$. Table 4 gives five sets of the typical values which are assigned respectively to the $\sigma_{Ps}$, $\sigma_{Tm}$, $P_s$ and $PWV$ in equations (10)~(12).

Table 4. Different typical values for $\sigma_{Ps}$, $\sigma_{Tm}$, $P_s$ and $PWV$

| Set of typical values | $\sigma_{Ps}$ [hPa] | $\sigma_{Tm}$ [K] | $P_s$ [hPa] | $PWV$ [mm] |
|---|---|---|---|---|
| (a) | 0.5 | 0 K ~ 7 K | 1013.25 | 50 |
| (b) | 0.5 | 0 K ~ 7 K | 850 | 50 |
| (c) | 0.5 | 0 K ~ 7 K | 1013.25 | 20 |
| (d) | 5 | 0 K ~ 7 K | 1013.25 | 50 |
| (e) | 5 | 0 K ~ 7 K | 1013.25 | 20 |

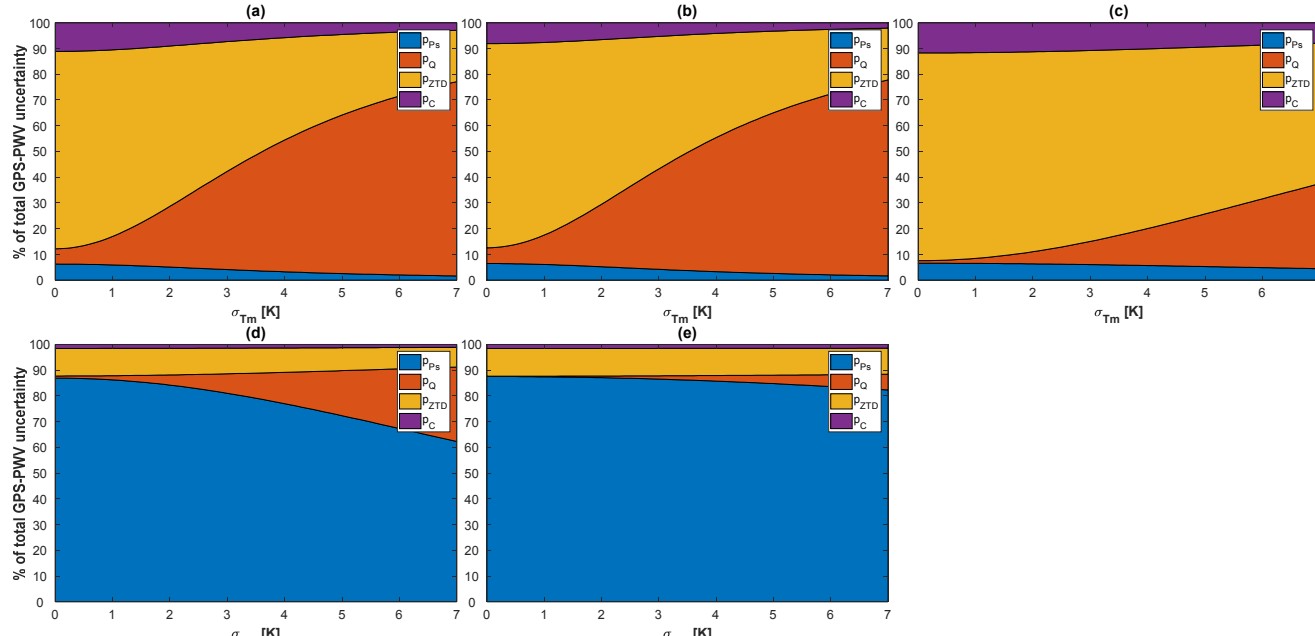

**Figure 10. Contributions of different terms to the total uncertainty of GPS-PWV with the different typical values shown in table 4**

The $\sigma_{Ps}$ equals 0.2 hPa in Ning et al. (2016), however we enlarged its typical value to 0.5 hPa in consideration of the possible worse performance of the surface barometers. In figure 10, we illustrated the contributions of the terms in equation (12) based on the assumptions (a) ~ (e) in table 4. Some variation features of the contributions of different terms can be found from the comparisons between different subplots:

(1) No significant difference exists between the figure 10(a) and 10(b). Because of the small value of $\sigma_c$ in equation (10), the $\sigma_{PWV}$ is not sensitive to the value of $P_s$. Meanwhile, the uncertainty associated with $\sigma_c$ contributes less than 10 % of the $\sigma_{PWV}$.

(2) With the typical values in table 4(a) and 4(b), a reduction of $\sigma_{Tm}$ can reduce the $p_Q$ significantly. For example in figure 10(a), the $p_Q$ accounts for 69.54 % with $\sigma_{Tm}$ = 6 K, and it declines to 38.19 % with $\sigma_{Tm}$ = 3 K.

(3) As figure 10(c) shows, the uncertainty associated with $\sigma_{ZTD}$ accounts for the main part of $\sigma_{PWV}$ when the values of $PWV$ and $\sigma_{Ps}$ are not high. With the typical values in table 4(c), the $p_{ZTD}$ can be up to 74.21 % with $\sigma_{Tm}$ = 3 K. And the $p_Q$, however, can drop from 26.76 % to 9.00 % as the $\sigma_{Tm}$ decreases from 6 K to 3 K. Although the $p_Q$ is not large under this situation, a smaller $\sigma_{Tm}$ can still reduce the contribution of $\sigma_Q$ to the $\sigma_{PWV}$.

(4) The uncertainty associated with $\sigma_{Ps}$ dominates the error budget of PWV when the $\sigma_{Ps}$ is large. In figure 10(d~e), the $p_{Ps}$ is over 80 % with $\sigma_{Tm}$ < 3 K and $\sigma_{Ps}$ = 5 hPa. In figure 10(d), the $p_Q$ increases from 7.55 % to 23.19 % as the $\sigma_{Tm}$ raises from 3 K to 6 K. However, in figure 10(e), the $p_Q$ only grows from 1.29 % to 4.61 % with the same variation of $\sigma_{Tm}$.

Theoretical analyses on $\sigma_{PWV}$ were also carried out at two representative stations. At the IGRA station No.42971 (20.25°

N 85.83° E, in India), the mean value of $PWV$ is 53.88 mm. The RMSEs of $T_{m\_Bevis}$, $T_{m\_LatR}$, $T_{m\_static}$, $T_{m\_varying}$, and $T_{m\_GPT2w}$ are

respectively 4.30 K, 3.15 K, 2.41 K, 1.93 K and 1.97 K. The $\sigma_{Tm}$ in equation (11) was replaced by the calculated RMSEs,

and the $p_{ZTD}$, $p_{Ps}$, $p_C$ and $p_Q$ were generated with two typical values, 0.5 hPa and 5 hPa, assigned to the $\sigma_{Ps}$. With

$\sigma_{Ps}$ = 0.5 hPa, the $p_C$ accounts for around 7 % while the $p_{Ps}$ accounts for around 4 % of the total $\sigma_{PWV}$. By using different

$T_m$ estimations, the variations of $p_C$ and $p_{Ps}$ are both within 4 %. However, the $p_Q$ varies more evidently. It accounts for

average 55.69 %, 40.77 %, 30.70 %, 23.53 %, and 24.11 % of the $\sigma_{PWV}$ respectively with the estimations of $T_{m\_Bevis}$, $T_{m\_LatR}$,

$T_{m\_static}$, $T_{m\_varying}$, and $T_{m\_GPT2w}$. The $p_{ZTD}$ raises with the reduction of $p_Q$, e.g. from 36.23 % of $T_{m\_Bevis}$ to 62.53 % of $T_{m\_varying}$.

On the other hand, with $\sigma_{Ps}$ = 5 hPa, the $p_{Ps}$ accounts for more than 75 % of the $\sigma_{PWV}$ while the $p_Q$ decreases from

14.21 % of $T_{m\_Bevis}$ to 3.9 % of $T_{m\_varying}$.

At another representative station, the IGRA station No.50557 (49.17° N 125.22° E, in Northeast China), the mean PWV

is only 12.17 mm. The RMSEs of $T_{m\_Bevis}$, $T_{m\_LatR}$, $T_{m\_static}$, $T_{m\_varying}$, and $T_{m\_GPT2w}$ are respectively 5.16 K, 3.94 K, 3.54 K, 2.99

K and 5.10 K. We can see that the accuracy of $T_m$ has been improved significantly. However, because of the low average value

of PWV, the $p_{ZTD}$ averagely contributes over 73.5 % of the $\sigma_{PWV}$ while the $p_Q$ averagely contributes less than 10.5 %

assuming $\sigma_{Ps}$ = 0.5 hPa and less than 1.5 % assuming $\sigma_{Ps}$ = 5 hPa. But such discussion only concerns the average values. In

fact, even at this station there are still some high values of PWV, for example at UTC 12:00 July 22th of 2016, the PWV

reached 48 mm. For the observations with high PWV, the improvement in the accuracy of $T_m$ can still exert significant positive

impact on the reduction of $p_Q$.

       It is worth mentioning that the uncertainty of ZHD may be underestimated in some situations. There are two reasons for

this. Firstly, the calculation of ZHD assumes that the water vapor is not contributed to the mass of the atmosphere. The ZHD

error introduced by this assumption is often negligible. But in some very wet regions, the mass of water vapor could produce

significant errors to the ZHD calculation. Secondly and more importantly, the error of $P_s$ in equation (1) can be very large

sometime. Small $\sigma_{Ps}$ is reasonable when the surface barometer is calibrated routinely and equipped together with the GPS

antenna. However, if there were significant height difference between the GPS antenna and the barometer, the error of ZHD

would increase significantly. Snajdrova et al.(2006) found that 10 m of height difference approximately causes a difference of

3 mm in the ZHD. On the other hand, $P_s$ can be generated from NWP data if there were no nearby barometer to GPS site. The

error of $P_s$ could be very large using this method (Means and Cayan, 2013; Jiang et al., 2016). In these cases, the GPS-PWV

error reduction due to the more precise $T_m$ estimation will be very limited.

**5.2 Impact of real $T_m$ estimation**

To study the impact of $T_m$ on the real GPS-PWV retrieval, we first downloaded GPS ZTD products (Byun and Bar-Sever,

2009) at 74 IGS sites in the year 2016 from the NASA Crustal Dynamics Data Information System (CDDIS) ftp address

(ftp://cddis.gsfc.nasa.gov/pub/gps/products/troposphere/zpd). These selected GPS sites were equipped with meteorological

sensors so that the surface pressure and temperature measurements could also be obtained. ZHD was calculated using equation

(1). It is subtracted from ZTD to obtain ZWD. Then, $T_m$ was generated with six approaches: the first five $T_m$ series were $T_{m\_Bevis}$,

$T_{m\_LatR}$, $T_{m\_static}$, $T_{m\_varying}$ , and $T_{m\_GPT2w}$. The sixth $T_m$ was integrated from the ERA-Interim profiles and interpolated to each

GPS site (Jiang et al., 2016; Wang et al., 2016). Finally, the GPS-PWV was generated from the ZWD and the six different $T_m$

estimates leading to over one hundred compared points for each GPS-PWV series. We denoted these GPS-PWV sets as

$PWV_{BTm}$, $PWV_{LTm}$, $PWV_{STm}$, $PWV_{VTm}$, $PWV_{GTm}$, and $PWV_{ETm}$. The only difference between these GPS-PWV estimations is

the $T_m$ estimation model; therefore, the impact of other errors is excluded.

The $T_m$ from ERA-Interim is believed to be the most accurate among our $T_m$ estimates at the selected GPS sites. We

therefore took the $PWV_{ETm}$ as reference values to assess the other PWV. The relative RMSEs of $PWV_{BTm}$, $PWV_{LTm}$, $PWV_{STm}$,

$PWV_{VTm}$ and $PWV_{GTm}$ at these selected stations were calculated and are illustrated in figure 11. The detailed statistics are

given in table 5. The mean relative error of all sites drops from 1.18 % of the $PWV_{BTm}$ to 0.91 % of the $PWV_{VTm}$. $PWV_{VTm}$

has the minimum mean relative errors at 51.35 % of the sites, while $PWV_{STm}$ is superior at 27.03 % of the sites. $PWV_{STm}$ and

$PWV_{VTm}$ obtain relative RMSE smaller than 1.0 % at 55 sites, while only 28 sites of $PWV_{BTm}$, 31 sites of $PWV_{LTm}$ and 22 sites

of $PWV_{GTm}$ perform similarly. For example, at ALIC site (23.67° S 133.89° E, in Australia), with a mean PWV of

approximately 23 mm, the relative RMSE dropped from 1.97 % of $PWV_{BTm}$ to 1.10 % of $PWV_{VTm}$. The time series of the

relative differences of $PWV_{BTm}$, $PWV_{LTm}$, $PWV_{STm}$, $PWV_{VTm}$, and $PWV_{GTm}$ are given in figure 12. We found that some relative

RMSEs could reduce more than 2 % from $PWV_{BTm}$ to $PWV_{VTm}$. Obviously, $PWV_{BTm}$ and $PWV_{LTm}$ have larger relative errors

throughout the year while the PWV differences are significantly larger only in the summer season (when the PWV values are

highest). Apparently, the $T_m$ variations in summer are not modeled well by both Bevis model and the latitude-related model.

$PWV_{STm}$ eliminate those large differences but still retain some residual errors, which are removed by more than 0.5 mm in

$PWV_{VTm}$. $PWV_{GTm}$ has some large errors during the period from May to July. All of these results demonstrate that our time-

varying model has precision advantages.

**Table 5: Statistics about the relative errors of different PWV retrievals**

| Statistics | $PWV_{BTm}$ | $PWV_{LTm}$ | $PWV_{STm}$ | $PWV_{VTm}$ | $PWV_{GPT2w}$ |
|---|---|---|---|---|---|
| Mean relative RMSE of all sites | 1.18 % | 1.12 % | 0.93 % | 0.91 % | 1.32 % |
| Number of sites with relative errors < 1.0 % | 28 | 31 | 55 | 55 | 22 |

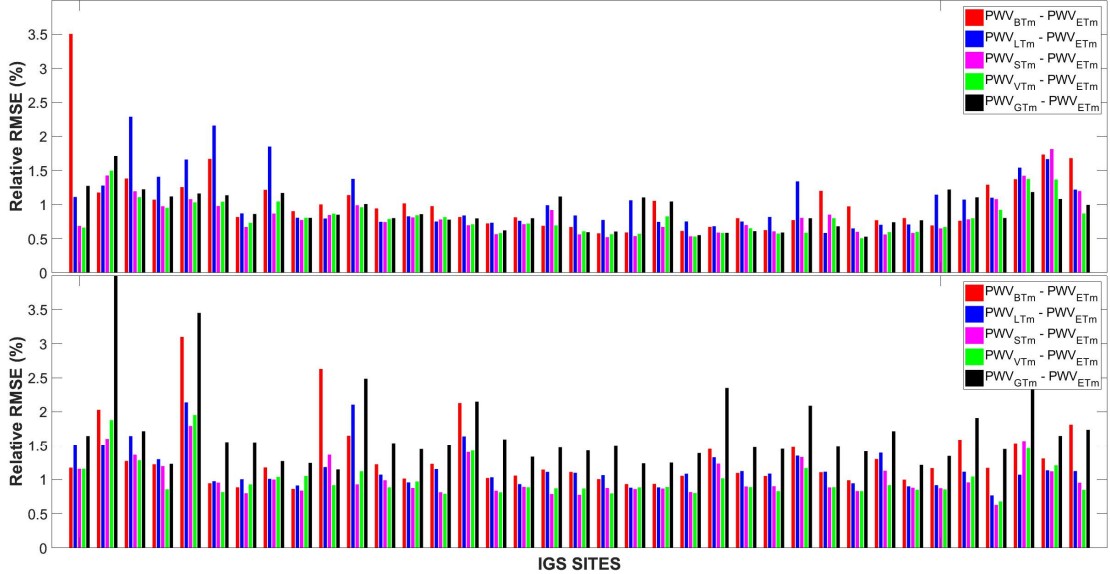

**Figure 11: Relative RMSEs of PWV_BTm, PWV_STm, PWV_VTm and PWV_GTm compared with PWV_ETm at 74 IGS stations in the year 2016**

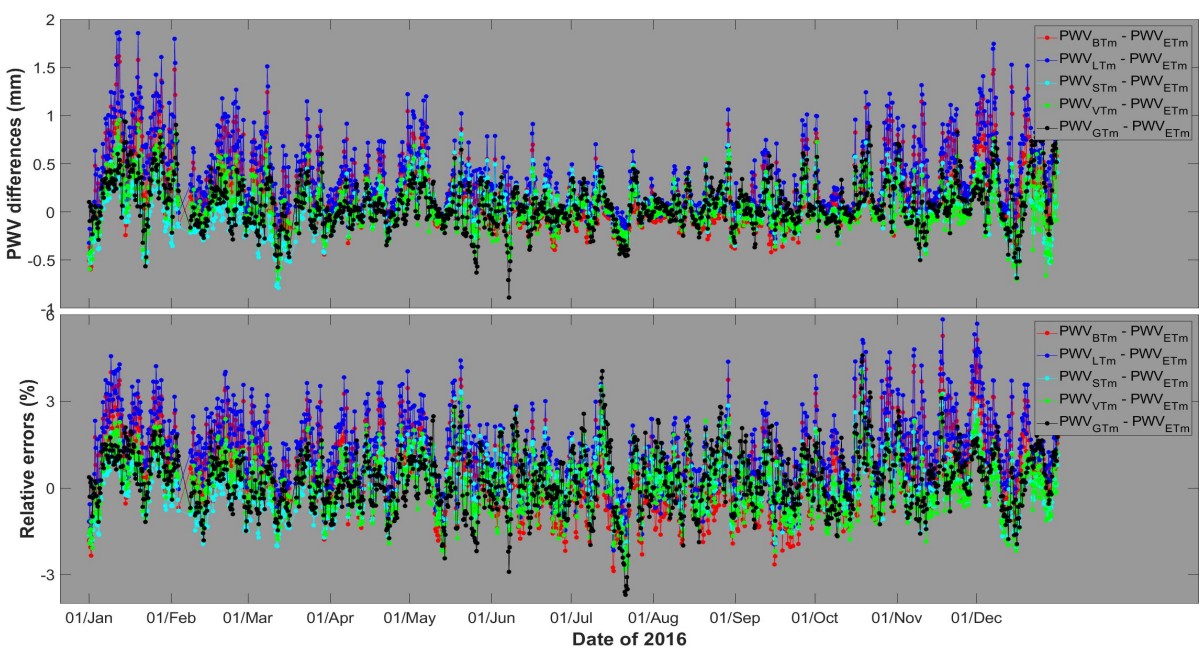

**Figure 12: (top) PWV differences and (bottom) relative differences of PWV_BTm, PWV_LTm, PWV_STm, PWV_VTM and PWV_GTm compared with PWV_ETm at the ALIC station in the year 2016. PWV unit is mm.**

### 5.3 Comparisons between GPS-PWV and radiosonde PWV

Among our selected 74 IGS sites, there are only 11 sites located within 5 km to a nearby IGRA radiosonde station. At these common stations, we generated PWV from the radiosonde data (PWV_RS) by adjusting the sounding profiles to the heights of IGS sites. It is worth noting that geoid undulation correction should be carried out on each IGS site geoid height (Jiang et al., 2016). Then, we compared PWV_BTm, PWV_LTm, PWV_STm, PWV_VTm, PWV_GTm, and PWV_ETm with PWV_RS. Figure 13 shows the statistics. The RMSEs of GPS-PWV are approximately 1~5 mm. Comparisons indicate that the RMSEs of different GPS-PWV retrievals are very close (differences < 0.2 mm) regardless of the applied $T_m$ sources at most of the selected sites. This

means that other errors (e.g. ZTD estimation errors or sounding sensors errors) instead of the $T_m$ make up the bulk of the

differences between the GPS-PWV and the radiosonde PWV. Actually, each sounding does not represent the vertical sounding

centered at the radiosonde site because of the complex path of the balloon. And GPS-PWV represents the averaged value of

the water vapor zenithal projection from all the slant signal paths during the observation period. Such differences can introduce

significant uncertainty to our comparisons. However, we still found obvious gaps between PWV at NRIL station (88.36° N

69.36° E, 4.1 km away from the IGRA station No.23078 in Russia). The RMSE decreases from 2.29 mm of $PWV_{BTm}$ to

1.84mm of $PWV_{VTm}$ and 1.42 mm of $PWV_{ETm}$. As shown in figure 14, the large PWV differences appear mainly from May to

September. During those five months, the mean GPS-PWV difference to $PWV_{RS}$ decreases by over 30 % from 2.52 mm of

$PWV_{BTm}$ to 1.67 mm of $PWV_{VTm}$, and the reductions of GPS-PWV error are mainly around 1~2 mm. This is attributed to the

wetter atmosphere in these months. As indicated by the uncertainty analysis in section 5.1, the improvement in the accuracy

of $T_m$ can be translated in more error reduction in the GPS-PWV retrieval with higher value of PWV.

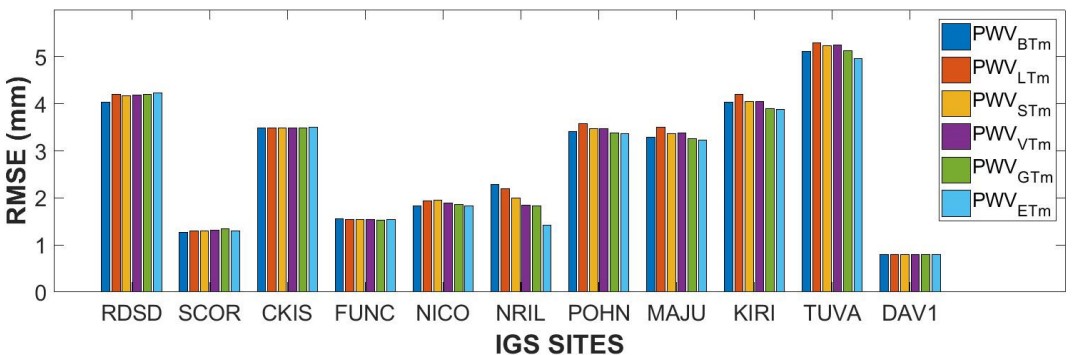

**Figure 13: RMSEs of the PWV$_{BTm}$, PWV$_{STm}$, PWV$_{VTm}$, PWV$_{GTm}$ and PWV$_{ETm}$ compared with the PWV$_{RS}$ at 11 IGS stations in 2016. PWV unit is mm.**

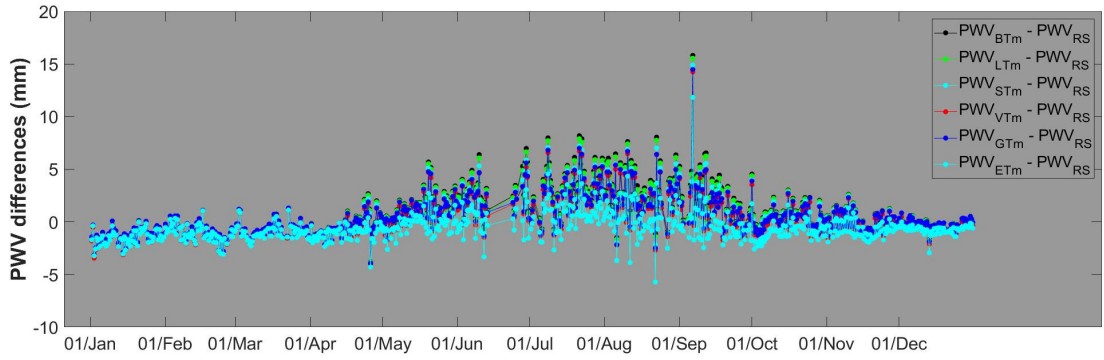

**Figure 14: PWV differences of the PWV$_{BTm}$, PWV$_{LTm}$, PWV$_{STm}$, PWV$_{VTm}$ , PWV$_{GTm}$ and PWV$_{ETm}$ compared with the PWV$_{RS}$ at NIRL station in the year 2016. PWV unit is mm.**

## 6. Summary and conclusion

We developed two global gridded $T_s$-$T_m$ models which are respectively static and time-varying with a spatial resolution

of 0.75° × 0.75°. The models are established by analyzing the ERA-Interim reanalysis datasets covering the year 2009~2012,

which indicated the significant spatial-temporal variations in $T_s$-$T_m$ relationship as well as the radiosondes covering the same period. The annual, semiannual, and diurnal variations in $T_s$-$T_m$ relationship are considered in the time-varying model. The time-varying global gridded $T_s$-$T_m$ model has a significant global precision advantage over the other global applied models, including the Bevis equation, the latitude-related model and the GPT2w model. Average RMSE of $T_m$ reduces by approximately 1 K. At over 90 % of the radiosonde sites, our time-varying model has RMSE smaller than 4 K, while the RMSE larger than 5 K nearly disappear. On the other hand, in the Bevis model or in the latitude-related model, there are more than 17 % of the radiosonde sites having RMSE larger than 5 K. Multiple statistical tests at the 5 % significance level identified the significant superiority of our varying model at more than 60 % of the radiosonde sites. Analyses at the specific stations demonstrate that the errors larger than 5 K in the estimated $T_m$ series can be eliminated by our varying $T_s$-$T_m$ model.

More precise $T_m$ estimation can reduce around 20 % of the uncertainty in the conversion factor $Q$ which maps GPS-ZWD to GPS-PWV, and the reduction can be even more than 50 % at some stations. The contribution of the uncertainty associated with $Q$ to the total GPS-PWV uncertainty also declines by using a more precise $T_m$ model. The reduction is related to the value of PWV and the uncertainty of surface pressure. With GPS-PWV higher than 50 mm, the uncertainty associated with $Q$ contributes more than 55 % of the uncertainty of GPS-PWV by using the Bevis equation and less than 25 % by using our varying $T_s$-$T_m$ model, assuming the ZTD and the surface pressure are measured accurately respectively with the uncertainties of 4 mm and 0.5 hPa. However, the uncertainty in ZTD or in surface pressure would dominate the error budget of GPS-PWV (> 70 %) if the value of GPS-PWV were small or the uncertainty of surface pressure were large. In these cases, the uncertainty associated with $Q$ only contributes around 10 % of the GPS-PWV uncertainty or even smaller. Taking the GPS-PWV using ERA-Interim $T_m$ estimates at 74 IGS sites as the references, we found that the GPS-PWV using our time-varying $T_s$-$T_m$ model obtained the minimum mean relative error at 51.35 % of the sites, while the GPS-PWV using the static gridded $T_s$-$T_m$ model is superior at only 27.03 % of the sites. The differences between GPS-PWV and radiosonde PWV are approximately 1~5 mm. And our varying $T_s$-$T_m$ model can reduce 30 % (around 1~2 mm) of the error in GPS-PWV retrieval with respect to the Bevis equation.

According to our experiments, we are confident that the time-varying global gridded $T_s$-$T_m$ models presented here will help us to retrieve GPS PWV more precisely and to study the precise PWV variations in high temporal resolution. Matlab array file consisting of the global gridded coefficients in our model, as well as codes to interpolate coefficients at any given location, are provided as the supplement of this study.

***Data sets***

Radiosonde data: ftp://ftp.ncdc.noaa.gov/pub/data/igra

ERA-Interim Project:    https://doi.org/10.5065/D6CR5RD9

GPS-ZTD Product: ftp://cddis.gsfc.nasa.gov/pub/gps/products/troposphere/zpd

Our model Supplement: https://www.atmos-meas-tech-discuss.net/amt-2018-67/amt-2018-67-supplement.zip

### *Acknowledgments*

*This study is supported by National Natural Science Foundation of China (No. 41604028), the Anhui Provincial Natural Science Foundation (No. 1708085QD83), and the Doctoral Research Start-up Funds Projects of Anhui University (No. J01001966). The authors thank European Centre for Medium-Range Weather Forecasts for providing*
*the ERA-Interim dataset. We also thank the National Centers for Environmental Information for the IGRA datasets and International GNSS Service for the GNSS troposphere products.*

### *Competing interests*

*The authors declare that they have no conflict of interest.*

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
