# Peer review of "Development of time-varying global gridded Ts-Tm model for precise GPS-PWV retrieval"

_Atmospheric Measurement Techniques, 2018_

## Referee Comment (RC1) · Anonymous Referee #2 · 12 May 2018

General Comment: The manuscript presented comprehensive investigation of the Ts-Tm relationship in both spatial and temporal domains using ECMWF reanalysis products. There are two interesting results: 1) Ts–Tm relationship has significant spatial and temporal variations; 2) the developed time-varying global gridded Ts–Tm equations perform better than the static model. The presentation of this investigation is well structured and written. I would like to recommend it for publication after the following modifications.

1 GPT2w model can also provide the Tm globally for real time applications. Authors should include this model in the comparisons. 2 What do the numbers in Figure 4 stand for? Are they results from the radiosonde? If so, you can also use the circles as shown in Figure 1. 3 Explain $m1$, $m2$, $n1$, $n2$ in equation (5) 4 When interpolating the Tm onto

the GNSS sites, did you consider the impact of height differences?

---

## Short Comment (SC1) · 23 Jul 2018

Dear Referee,

Thank you very much for your careful review and positive response to our manuscript. We also appreciate the good suggestions provided by you. Please find our responses to your comments and revised manuscript in the supplement files. Best regards,

The authors

Please also note the supplement to this comment: https://www.atmos-meas-tech-discuss.net/amt-2018-67/amt-2018-67-SC1-supplement.zip

---

## Referee Comment (RC2) · Anonymous Referee #3 · 31 Aug 2018

**Review of "Development of time-varying global gridded Ts-Tm model for precise GPS-PWV retrieval", by Jiang, Ye, Lu, Liu, Chen and Wu**

**General comments**

This is a nice article that I enjoyed reading.

The authors use ERA Interim data for the years 2009 to 2012 to determine surface temperature, Ts, and weighted mean temparature, Tm, with a time resolution of 6 hours for a global grid with resolution 0.75 x 0.75 degrees. Based on this, static, linear Ts-Tm relations are deduced for each gridpoint, as well as linear Ts-Tm relations that on top include annual, semiannual and diurnial variations (modelled with cosines). The global properties of these data sets are shown.

Secondly the performance of these two models, as well as of the Bevis formula and a latitude related model is compared to Tm data from 758 radiosonde stations for the year 2016. It is concluded convincingly that with a few exceptions the ERA Interim gridded Ts-Tm model including annual, semiannual and diurnial variations is superior to the other sources of Tm in the comparison.

In connection with publication of the article the authors will make available to the community their model.

The material deserves publication.

There are some unclear points, figures that can be improved, etc., calling for a revision of the article.

The English is not good. When making their revision the authors should have a person with good command of English in scientific writing help them.

**Some aspects to consider in a revision**
Consider using the term integrated water vapour, IWV, instead of PWV.

Section around line 65: Is it well established that global empirical Tm models without Ts components are less good? If so give some references. If not, include some of them in your comparisons of different sources of global Tm. We want to know whether your dataset is the best global set for Tm estimation around for the moment, or just better than other Ts-Tm based sets.

Notice that in numerical weather prediction one in general uses GPS ZTD, not PWV. PWV is important for climate monitoring, and for meteorologists doing weather forecasting combining information from weataher prediction models and observations.

around line 100: Did you take into consideration that water vapour pressure (and density) varies approximately exponentially with height when doing the integral in eq. 4. It is not likely to have a large impact, but the fewer RS levels you have access to, the larger the effect. 5 levels is not a lot.

around line 120: "geoid height" should be "geometric height". The reference surface doesn't matter for the integral.

around line 150: There are several places where text is not properly separated, here E180. etc., text of

figure 3 is another example.

Figure 1. It is hard to see properly the RS circles. Consider making the figure little bit larger, and draw a thin black line around the RS circles, in order that one can see them also where they agree with ERA.

At many places in the text and in the figures units are missing.

Figure 6: Are the colors plotted in a particular order, such that for example large rms will be plotted on top of small rms? If so, do a check that plotting in the opposite order yield almost similar plots. Otherwise enlarge.

Regarding the RS ERA comparisons. Is anything done to handle altitude offsets between RS surface and ERA surface?

Around line 280.  At 10.82 % of the sites inclusion of the time variations in ERAI resulted in a poorer results. That indicates ERAI has particular problems at these locations. If you plot them on a map, do you see any systematics in their location?

Around line 310. The pressure used to determine ZHD should be the pressure at the GNSS antenna level, not surface pressure. Did you do something to correct for height offsets, or is the barometer installed at the same altitude as the GNSS antenna at these locations?
Similarly the Tm integral should in principle run from the antenna level and up, not from the surface. In almost all cases that is not likely to create problems, but there will be locations where the difference between the surface altitude of ERA and the altitude of a GNSS site is huge. I'm not familiar with the location of IGS sites, but for GNSS reference sites in general the altitude difference can be more than 1000 m between a GNSS site and an NWP model with higher horisontal resolution than ERAI

Around line 385: It would nice if in the final article you could add an extra line with a link to your dataset. It seems very useful to many people :-)

---

## Referee Comment (RC3) · D. Adams (Referee) · 21 Sep 2018

Review of Peng et al. 2018
David K. Adams (dave.k.adams@gmail.com)

Recommendation:  Major Revisions

**General Comments**
Firstly, this paper is in terrible need of proofreading by a native English speaker.
The paper is very poorly written and is nearly incomprehensible is some parts.
Correcting grammar errors is not the responsibility of the reviewer; this paper should have been checked beforehand.    After line 170, I have stopped correcting these grammar errors and I just focus on content.

The subject of the the Bevis and other models has been examined many, many times, globally and regionally.  See the work of June Wang (SUNY) and for example Luiz Sapucci (Brazil).  There has to be a very strong motivation to continue this type of work.  Improvements of 1 or 2mm in PWV are not very impressive and hardly seem worthwhile considering all of the other inherent errors in the GPS PWV method itself, errors in radiosondes, errors in ERA Interim data and other reanalysis products (which may not be independent of radiosonde in the first place).

The authors need to better justify why these tiny improvements given the work required are really advantageous.  More importantly, errors due to surface pressure are much larger.  The height of the antenna relative to the surface pressure measurement will introduce a much larger error than what you present here with T_m models.   You should do an analysis of these errors.  How large are the errors associated with a reasonable error or 5mb in surface pressure?   Likewise, the assumption of ZHD representing the mass of the atmosphere (excluding water vapor) is erroneous.  What are the errors associated with this assumption?  Compare the Amazon to a desert region to assess this error somehow.

Add these analyses to the study and it would be improved.

**Specific Comments.**
Check your spacing on the in-text citations.

Abstract
Line 10 Write  "In near real-time GPS-PWV retrievals,"

Line 13  Write " without data smoothing"

Line 14 "Then static and time-varying global gridded T s –T m"
I am not sure what you mean here by "Then"

Line 16  Write "have prominent advantages over other"

 Line 17   Write "Large biases in Bevis' equation or  in latitude-related linear models at a considerable number of stations"

Line 18  Write  "Multiple statistical tests at the  5% significance level"

Line 29-30.  This sentence is awkward, rewrite.

Line  31 Write "such as radiosondes and water vapor radiometers,"

Line 36.  Slant water vapor(SWV) are not "widely used"

Line 37-38   I don´t know what you are trying to say here.

Line 44  Write "The mapping function..."

Line 49.  Ps  also has a significant effect on PWV calculated from GPS signal delays, because it represents the mass of the dry atmosphere.

Line 54  Write " The integration of"

Line 58-60  Rewrite this sentence, it is difficult to follow.

Line 66 Write " can be lost without the constraint of real data."

Line 68 Write "numerical weather prediction."

Line 80.  "there exist large differences between the oceanic and terrestrial atmospheric properties."
This depends.  At the surface layer and boundary layer, yes, but about the boundary layer it is less clear.
You should specify what you mean here.

Line 81  Write "oceanic regions" not sea regions

Line 85 "however is statistic and the estimated T m residuals due to time variations are not fixed (Yao et al., 2014a)."
I don´t understand what you are trying to say here.

Line 87 Write "spatial smoothing of the data"

Line 101 What do you mean specifically "atmospheric top" (i.e. top of the atmosphere).  Give a value.

Line 105 Write "We employed radiosonde data from  the Integrated"

Line 106.  You should show PWV sensitivity to surface pressure.  Using Bevis´ model, the sensivity if about 3 times greater to surface pressure than to Ts.

Line 107  Write "may be"

Line 109 Surface **observations** must be available, and top profile level should not be lower than 300 hPa standard level.
Oftentimes, the first level of a radiosonde has erroneous data, over influenced by surface conditions. You should quality check the first level of the sounding for bad temperature and humidity data.

LIne 111.  Much greater the 5 levels 1000mb to 300mb is necessary.  Maybe a more stringent criterion is needed.  Maybe 10 levels, you should check the sensitivity of your results to this assumption.

Line 113. "Profile data including same elements are usually provided by NWP products at certain vertical levels."
It is not clear what you mean by same elements. I assume you mean the same variables.
You should also say what the cone of representation of the GPS is, so one can consider that with respect to the "vertical" measurements of the radiosondes.

Line 116. ERA Interim humidity products can be awful, particularly in regions where little observational data exist. ERA-5 somehow is better, not sure, how they managed this.

Line 120 Write "However, water vapor is solely concentrated in the troposphere, and most of it, specifically within the first 3 kilometers above sea-level."

Line 125 "to the height replacement will extremely approximate to zero."
This doesn´t make sense, rewrite.

Line 137 Write "We first carried out a linear regression analysis"

Line 140 Correct this sentence.

Line 145 Write "It is evident that T m varies"

Line 148 This sentence makes no sense.

Line 169 Write "Since the T s -T m relationship has"

Line 236. Remember that the radiosondes are incorporated/assimilated into reanalysis products, so the data are not independent. So you may not be correctly capturing real errors and biases in the data.

Line 314. Remember, there is an inherent error in the assumption of the calculation of ZHD, which assumes that the water vapor is not contributed to the mass of the atmosphere. This error may not be important but should be evaluated for very wet regions (e.g, the Amazon) and very dry regions.

Line "Because the T m from ERA-Interim is believed to be the most accurate"
This is a very strong statement. You need to provide evidence considering the purpose of this paper is entirely dependent on the quality of the observations. As I said before these data are not necessarily independent of the radiosondes and their humidity data in regions with few observations can be awful.

The errors that you present as a function of T_m as tiny. I am perfectly happy to work with data with only of few percentage error. Maybe a 10% change is worth noting, but I am not too stressed about errors of these sizes, particularly considering all the assumption going in to the calculations of PWV from GPS ZTD.
Reference
Sapucci, L.F., 2014: Evaluation of Modeling Water-Vapor-Weighted Mean Tropospheric Temperature for GNSS-Integrated Water Vapor Estimates in Brazil. *J. Appl. Meteor. Climatol.,* **53**, 715–730, https://doi.org/10.1175/JAMC-D-13-048.1

---

## Author Comment (AC3) · 1 Nov 2018

Dear David Adams,
Thank you for your careful review and suggestions to our manuscript. Our responses to your comments and the revised manuscript are compressed in the supplement file.
Best regards,
the authors

Please also note the supplement to this comment:
https://www.atmos-meas-tech-discuss.net/amt-2018-67/amt-2018-67-AC3-supplement.zip

---

## Author Response (AR1)

Dear reviewers,

Thank you for your careful reviews and suggestions. Followings are our responses to your comments. Please note that the line numbers in our responses are the numbers in the revised manuscript using track change.

**Response to Anonymous Reviewer #2:**

1. GPT2w model can also provide the Tm globally for real time applications. Authors should include this model in the comparisons

**Response:** We have included Tm estimations from GPT2w model in all of our comparisons. And the results still indicate that our Ts-Tm model has accuracy advantages over other Tm estimation models. Detail statistics are included in the revised manuscript. (Line 452~775)

2. What do the numbers in Figure 4 stand for? Are they results from the radiosonde? If so, you can also use the circles as shown in Figure 1.

**Response:** The numbers in figure 4 are the contour values. In static Ts-Tm model Tm = $a$*Ts + $b$, $a$ stands for slope constant and b stands for intercept constant. We estimated $a$ and $b$ at each gird node of ERA-Interim data. In figure 4, top figure is the global color contour of $a$, middle figure is the global color contour of $b$, and bottom figure is the global color contour of Ts-Tm model RMSE.

3. Explain m1, m2, n1, n2 in equation (5).

**Response:** $(m_1, m_2)$, $(n_1, n_2)$ and $(p_1, p_2)$ are the fitting coefficients of formula (5) items, and these coefficients can indicate amplitudes of annual, semiannual and diurnal variations in our Ts-Tm models. We have included these explains in the revised manuscript (Line 431~432).

4. When interpolating the Tm onto the GNSS sites, did you consider the impact of height differences?

**Response:** We have considered two types of height differences in our interpolations:

(a) The difference between geodetic height (applied by GNSS) and geopotential height (applied by NWP). A GNSS site's geodetic height is converted to altitude height using EGM2008 model, and the difference between altitude height and geopotential height is neglected as we done in our another study(Jiang et al., 2016);

(b) The height differences between GNSS sites and NWP levels. For a GNSS site, we estimated the Tm at its four neighbor gird nodes in ERA-Interim data and then horizontally interpolated them onto GNSS site's location. At each gird node, the Tm integral geopotential height range is from GNSS site's geopotential height to 1 hPa level's height. The pressure, temperature and humidity at GNSS site's geopotential height is interpolated from its upper and lower ERA-Interim levels.

**Reference:**

Jiang, P., Ye, S. R., Chen, D. Z., Liu, Y. Y., and Xia, P. F.: Retrieving Precipitable Water Vapor Data Using GPS Zenith Delays and Global Reanalysis Data in China, Remote Sensing, 8, 10.3390/rs8050389, 2016.

**Response to Anonymous Reviewer #3:**

We have turned to a language editing service when prepared the revised manuscript. And the editors tell us that they will also offer English language copy-editing for final revised accepted papers. Followings are our responses to your comments:

1. Consider using the term integrated water vapour, IWV, instead of PWV.

**Response:** In our opinion, IWV has the same meaning as PWV. And "GPS PWV" seems to be used together more widely in many articles. So we choose the term "PWV" rather than "IWV".

2. Section around line 65: Is it well established that global empirical Tm models without Ts components are less good? If so give some references. If not, include some of them in your comparisons of different sources of global Tm. We want to know whether your dataset is the best global set for Tm estimation around for the moment, or just better than other Ts-Tm based sets.

**Response:** We have included Tm estimations from GPT2w model in all of our comparisons. GPT2w model is a global empirical Tm models newly developed by Bohm in 2015 (Bohm et al., 2015). And the results still indicate that our Ts-Tm model has accuracy advantages over other Tm estimation models. Detail statistics are included in the revised manuscript. (Line 452~775)

3. Notice that in numerical weather prediction one in general uses GPS ZTD, not PWV. PWV is important for climate monitoring, and for meteorologists doing weather forecasting combining information from weather prediction models and observations.

**Response:** Thank you for your kind suggestion. We have modified our expressions about the use of GPS PWV in weather prediction. (Line 166)

4. around line 100: Did you take into consideration that water vapour pressure (and density) varies approximately exponentially with height when doing the integral in eq. 4. It is not likely to have a large impact, but the fewer RS levels you have access to, the larger the effect. 5 levels is not a lot.

**Response:** Yes, we have considered the temperature and water vapor pressure's variations with height. Between two neighbor RS observation heights, we calculated the water vapor pressure at the middle height for equation (4) by exponential interpolation of the two RS heights' water vapor pressure, while we estimated the temperature by linear interpolation. We have added some explanations in our revised manuscript. (Line 278~279)

4. around line 120: "geoid height" should be "geometric height". The reference surface doesn't

matter for the integral.

**Response:** Revisions have been made. (Line 312)

5. around line 150: There are several places where text is not properly separated, here E180. etc., text of figure 3 is another example.

**Response:** Revisions have been made. (Line 361, 394, 663, 665, and 753)

6. Figure 1. It is hard to see properly the RS circles. Consider making the figure little bit larger, and draw a thin black line around the RS circles, in order that one can see them also where they agree with ERA.

**Response:** Figure 1 have been plotted bigger. Actually our figures are vector graphs, so readers can zoom in on them to see the plots more clearly.

7. At many places in the text and in the figures units are missing.

**Response:** Revisions have been made.(Figure 2,3,4,5,6,7,9,10,11,12, and 13)

8. Figure 6: Are the colors plotted in a particular order, such that for example large rms will be plotted on top of small rms? If so, do a check that plotting in the opposite order yield almost similar plots. Otherwise enlarge.

**Response:** We have enlarged our plots in figure 6 to ensure that no color point is covered by others.

9. Figure 6: Regarding the RS ERA comparisons. Is anything done to handle altitude offsets between RS surface and ERA surface?

**Response:** In Equation (5), which is the time-varying Ts-Tm model, there are eight coefficients ($a$, $b$, $m_1$, $m_2$, $n_1$, $n_2$, $p_1$, $p_2$) which are estimated at each ERA-I grid node. In figure 6, we evaluated the performance of equation (5) at each RS site. Considering the horizontal offsets between RS sites and ERA-I grid nodes, we obtained the eight coefficients ($a$, $b$, $m_1$, $m_2$, $n_1$, $n_2$, $p_1$, $p_2$) at each RS

location by horizontally interpolated the ones of RS site's four neighbor ERA-I grid nodes. However, we think that the height differences between RS sites and it neighbor ERA-I surfaces have little impact on our Ts-Tm models (not Tm estimations). Such impact on Ts-Tm model can be compensated by the Ts input, which should be changed with altitude.

10. Around line 280. At 10.82 % of the sites inclusion of the time variations in ERAI resulted in a poorer results. That indicates ERAI has particular problems at these locations. If you plot them on a map, do you see any systematics in their location?

**Response:** GPT2w model has been added in our comparisons and we modified our results in the revised manuscript.

Unfortunately we found that there is no obvious characteristic in the distribution of RS site with poorer results. The reasons for such poorer results seems to be complicated and need specific study in the future.

11. Around line 310. The pressure used to determine ZHD should be the pressure at the GNSS antenna level, not surface pressure. Did you do something to correct for height offsets, or is the barometer installed at the same altitude as the GNSS antenna at these locations?

**Response:** The air pressure for each ZHD calculation was measured by the barometer equipped together with GNSS antenna at the GNSS site. Their locations are considered to be the same. (Line 694~696)

12. Similarly the Tm integral should in principle run from the antenna level and up, not from the surface. In almost all cases that is not likely to create problems, but there will be locations where the difference between the surface altitude of ERA and the altitude of a GNSS site is huge. I'm not familiar with the location of IGS sites, but for GNSS reference sites in general the altitude difference

can be more than 1000 m between a GNSS site and an NWP model with higher horizontal resolution than ERAI.

**Response:** We have considered such height differences in our comparisons. We interpolated (or extrapolated) ERA-I profile to GNSS antenna's location, and then started the Tm integral from GNSS antenna's altitude.

13. Around line 385: It would nice if in the final article you could add an extra line with a link to your dataset. It seems very useful to many people.

**Response:** A link to our model has been added. (Line 833).

**Reference:**

Bohm, J., Moller, G., Schindelegger, M., Pain, G., and Weber, R.: Development of an improved empirical model for slant delays in the troposphere (GPT2w), Gps Solutions, 19, 433-441, 10.1007/s10291-014-0403-7, 2015.

**Response to David Adams:**

**General Comments**

1. Firstly, this paper is in terrible need of proofreading by a native English speaker. The paper is very poorly written and is nearly incomprehensible is some parts. Correcting grammar errors is not the responsibility of the reviewer; this paper should have been checked beforehand. After line 170, I have stopped correcting these grammar errors and I just focus on content.

**Response:** Thank you very much for your careful corrections. We have turned to a language editing service when prepared the revised manuscript. And the editors tell us that they will also offer English language copy-editing for final revised accepted papers.

2. The subject of the Bevis and other models has been examined many, many times, globally and

regionally. See the work of June Wang (SUNY) and for example Luiz Sapucci (Brazil). There has to be a very strong motivation to continue this type of work. Improvements of 1 or 2mm in PWV are not very impressive and hardly seem worthwhile considering all of the other inherent errors in the GPS PWV method itself, errors in radiosondes, errors in ERA Interim data and other reanalysis products (which may not be independent of radiosonde in the first place).The authors need to better justify why these tiny improvements given the work required are really advantageous. More importantly, errors due to surface pressure are much larger. The height of the antenna relative to the surface pressure measurement will introduce a much larger error than what you present here with Tm models. You should do an analysis of these errors. How large are the errors associated with a reasonable error or 5mb in surface pressure? Likewise, the assumption of ZHD representing the mass of the atmosphere (excluding water vapor) is erroneous. What are the errors associated with this assumption? Compare the Amazon to a desert region to assess this error somehow.

**Response:** The uncertainty of the PWV estimated from GNSS observations has been discussed comprehensively in detail by Ning's research in 2016 (Ning et al., 2016). In his study, the uncertainties in ZTD estimation, ZHD estimation and the conversion factor Q from ZWD to PWV were analyzed and their contributions to the total uncertainty of GPS PWV were evaluated. However, Ning's study assumed the Tm were obtained from NWP models. We estimated the Tm from surface air temperature Ts. So we replaced the uncertainty of Tm by our statistical results. Considering the uncertainty of Tm is related to the uncertainty of conversion factor Q, we calculated the percentages of total GPS PWV's uncertainty due to the errors in Q at radiosonde stations over the world. We found that the contribution of Q's uncertainty to the total GPS PWV's uncertainty has been dropped significantly by using more precise Ts-Tm model. The experiments and results are shown in line

613~652 of the revised manuscript.

The differences between the statistical results of various Tm models showed in table 2 seemed to be tiny. Actually the improvements due to the application of more precise Tm model were very evident at a considerable number of sites (or during wet seasons) as showed in figure 8/9. But such significant improvements were smoothed in the statistical results with respect to all sites and all seasons.

**Specific Comments.**

1. Check your spacing on the in-text citations.

Line 10 Write "In near real-time GPS-PWV retrievals"

Line 13 Write " without data smoothing",

Line 14 "Then static and time-varying global gridded T s –T m",

Line 16 Write "have prominent advantages over other",

Line 17 Write "Large biases in Bevis' equation or in latitude-related linear models at a considerable number of stations",

Line 18 Write "Multiple statistical tests at the 5% significance level",

Line 29-30. This sentence is awkward, rewrite,

Line 31 Write "such as radiosondes and water vapor radiometers,",

Line 44 Write "The mapping function..."

Line 54 Write " The integration of"

Line 66 Write " can be lost without the constraint of real data.",

Line 68 Write "numerical weather prediction.",

Line 81 Write "oceanic regions" not sea regions,

Line 87 Write "spatial smoothing of the data",

Line 105 Write "We employed radiosonde data from the Integrated",

Line 107 Write "may be",

Line 120 Write "However, water vapor is solely concentrated in the troposphere, and most of it,

specifically within the first 3 kilometers above sea-level.",

Line 137 Write "We first carried out a linear regression analysis",

Line 145 Write "It is evident that T m varies",

Line 169 Write "Since the T s -T m relationship has"

**Response:** Grammar revisions have been made in the revised manuscript.

2. Line 36. Slant water vapor(SWV) are not "widely used"

**Response:** We deleted SWV in the revised manuscript.

3. Line 37-38 I don´t know what you are trying to say here.

**Response:** We rewrite the sentences in the revised manuscript. (Line 59~60)

4. Ps also has a significant effect on PWV calculated from GPS signal delays, because it represents

the mass of the dry atmosphere.

And Line 106. You should show PWV sensitivity to surface pressure. Using Bevis´ model, the

sensivity if about 3 times greater to surface pressure than to Ts.

**Response:** As our response to general comments #2, the impact of Ps was considered in the newly

added experiment.

5. Line 58-60 Rewrite this sentence, it is difficult to follow.

**Response:** We rewrite this sentence to "However, another data source, radiosonde data, has low spatial and temporal resolution. At most of the radiosonde sites, sounding balloons are daily cast at 00:00 UTC and 12:00 UTC. Furthermore, a large amount of GPS stations are not located close enough to the radio sounding sites." (Line 140-142)

6. Line 80. "there exist large differences between the oceanic and terrestrial atmospheric properties." This depends. At the surface layer and boundary layer, yes, but about the boundary layer it is less clear. You should specify what you mean here.

**Response:** We rewrite this sentence to "Significant differences exist between the oceanic and terrestrial atmospheric properties, especially at the surface layer and boundary layer. The change of Ts from land to ocean may be very different from that of Tm." (line 179~180)

7. Line 85 "however is statistic and the estimated T m residuals due to time variations are not fixed (Yao et al., 2014a)." I don´t understand what you are trying to say here.

**Response:** We rewrite this sentence to "However, the Ts-Tm relationship has time variations and can produce residuals in the static Tm estimations (Yao et al., 2014a). Such residuals are not fixed in Lan's model." (Line 184~185).

8. Line 101 What do you mean specifically "atmospheric top" (i.e. top of the atmosphere). Give a

value.

**Response:** We rewrite this sentence to "The integral intervals are from earth surface to top level. The height of the top level depends on the data sources we employed, which will be shown in section 2.2." (Line 279~281)

9. Surface observations must be available, and top profile level should not be lower than 300 hPa standard level. Oftentimes, the first level of a radiosonde has erroneous data, over influenced by surface conditions. You should quality check the first level of the sounding for bad temperature and humidity data.

**Response:** The IGRA data providers declare that NCEI scientists have applied a comprehensive set of quality control procedures to the data to remove gross errors ([https://www.ncdc.noaa.gov/data-access/weather-balloon/integrated-global-radiosonde-archive](https://www.ncdc.noaa.gov/data-access/weather-balloon/integrated-global-radiosonde-archive)). We also performed a quality check to remove the abnormal temperature and humidity data.

10. Much greater the 5 levels 1000mb to 300mb is necessary. Maybe a more stringent criterion is needed. Maybe 10 levels, you should check the sensitivity of your results to this assumption.

**Response:** We adjusted the criterion to 10 levels (line 291). Furthermore, we considered the temperature and water vapor pressure's variations with height. Between two neighbor RS observation heights, we calculated the water vapor pressure at the middle height for equation (4) by exponential interpolation of the two RS heights' water vapor pressure, while we estimated the temperature by linear interpolation. We added some explanations in our revised manuscript (line 278~279). According to our experiments, the Tm differences due to the number of radiosonde levels

are smaller than 0.55 K which are significantly smaller than the Tm's uncertainty shown in table 2. So in our opinion, it is reasonable to employ these radiosonde data to evaluate the errors of Tm estimation model.

The number of radiosonde stations in our comparisons decreased to 723 due to the change of such criterion. Therefore some statistical results changed in figure 6~8 and table 2~3.

11. "Profile data including same elements are usually provided by NWP products at certain vertical levels". It is not clear what you mean by same elements. I assume you mean the same variables. You should also say what the cone of representation of the GPS is, so one can consider that with respect to the "vertical" measurements of the radiosondes.

**Response:** We have realized the word "same elements" is not proper for what we tried to express. We meant the temperature, relative humidity and geopotential which can be used to calculate Tm. However, there are much more other variables in ERA-Interim, so we rewrite this sentences to "Profile data are usually provided by NWP products at certain vertical levels." In the comparisons between GPS PWV and radiosonde PWV, we described the differences of the physical meaning between the different PWV results. (Line 750~753)

12. Line 125 "to the height replacement will extremely approximate to zero." This doesn´t make sense, rewrite.

**Response:** We rewrite this sentence to "The Tm value nearly has no change after such height replacement" (line 317~318).

13. Line 140 Correct this sentence.

**Response:** We rewrite this sentences to "Our analyses also indicated that the correlation coefficient between Ts and Tm is generally related to the latitude. The same conclusion has been drawn in other studies (Yao et al., 2014b)."(Line 331~332).

14. Line 148 This sentence makes no sense.

**Response:** We realized that there is no experiment to support such guess, therefore this sentence is deleted in the revised manuscript.

15. Remember that the radiosondes are incorporated/assimilated into reanalysis products, so the data are not independent. So you may not be correctly capturing real errors and biases in the data.

**Response:** As described in line 457~459 of the revised manuscript, our model was developed from the NWP data covering the period from 2009~2012. And the radiosonde data used for the assessments in section 4.3 were observed in 2016. Therefore the two datasets are independent. Of course in both NWP and RS datasets, there may exist common bias due to the errors of sounding sensors. However the sounding data is oftentimes more precise than Ts-Tm model, so we think it is reasonable to evaluate Tm estimation model using radio sounding data.

16. Remember, there is an inherent error in the assumption of the calculation of ZHD, which assumes that the water vapor is not contributed to the mass of the atmosphere. This error may not be important but should be evaluated for very wet regions (e.g, the Amazon) and very dry regions.

**Response:** Experiments have been added as described in the response to the general comment #2.

17. "Because the Tm from ERA-Interim is believed to be the most accurate" This is a very strong statement. You need to provide evidence considering the purpose of this paper is entirely dependent on the quality of the observations. As I said before these data are not necessarily independent of the radiosondes and their humidity data in regions with few observations can be awful.

**Response:** We rewrite this sentence to "The Tm from ERA-Interim is believed to be the most accurate among our Tm estimates at the selected GPS sites" (line 712). We assumed this based on some studies which claimed that the NWP datasets were good choices for global Tm estimation (Wang et al., 2005; Wang et al., 2016). We also repeated those comparisons between NWP and radiosonde datasets and concluded the similar accuracy. Although the ERA-Interim data has assimilated the data from various satellites and radiosonde, we found that the ERA-Interim Tm estimates do not agree well with the real observations in some regions. These large errors have impacts on our Ts-Tm models as we described in line 670~673 of the revised manuscript.

**Reference:**

Ning, T., Wang, J., Elgered, G., Dick, G., Wickert, J., Bradke, M., Sommer, M., Querel, R., and Smale, D.: The uncertainty of the atmospheric integrated water vapour estimated from GNSS observations, Atmos. Meas. Tech., 9, 79-92, 10.5194/amt-9-79-2016, 2016.
Wang, J. H., Zhang, L. Y., and Dai, A.: Global estimates of water-vapor-weighted mean temperature of the atmosphere for GPS applications, Journal of Geophysical Research-Atmospheres, 110, 10.1029/2005jd006215, 2005.
Wang, X. M., Zhang, K. F., Wu, S. Q., Fan, S. J., and Cheng, Y. Y.: Water vapor-weighted mean temperature and its impact on the determination of precipitable water vapor and its linear trend, Journal Of Geophysical Research-Atmospheres, 121, 833-852, 10.1002/2015jd024181, 2016.

---

## Referee Report (RR1)

**Re-review of "Development of time-varying global gridded Ts-Tm model for precise GPS-PWV retrieval", by Jiang, Ye, Lu, Liu, Chen and Wu**

**General comments**
In my report on the first version of the manuscript I concluded that it contained interesting material that deserved publication with minor revision of the scientific part and heavy revision of the presentation, due to poor English and other errors.

Unfortunately the situation is still the same.

Scientifically an important improvement has been made by inclusion of the GPT2w model among the global Tm providing models to which the authors new model is compared. This said under the assumption it is the 1 by 1 degree resolution GPT2w version that is used, if it is the 5 by 5 degree, the comparison should be redone.

Language wise the manuscript has been improved a bit, but it is still far from adequate. According to the authors the manuscript has passed a language editing service, but there are still way more errors than what referees and editors can be expected to deal with.

I notice with surprise that the manuscript is still full of occurences where a space is missing in the text (such as writing "precipitable water vapor(PWV)" instead of "precipitable water vapor (PWV)". It is particularly found in connection with citations. It does not require English language skills to cure that, just a modest effort. When there are six authors, I expect the number of such mistakes to be very close to zero in a second manuscript, when they were notified about the problem in a report on the first manuscript. It is a harsh comment to make, but to me high number of such errors in the second manuscript it is a signal that so far the authors have not taken proper presention of their material seriously.

I'm sorry to say, but the manuscript must be thoroughly revised to improve the language and other writing errors.

**Detailed comments (far from exhaustive regarding presentation errors)**
Remove the paragraph in lines 53 to 55, including equation 5. You do a better error budget on page 20.

Lines 68 to 71 contain a statement about Tm models not including current observations not being accurate enough. You must either give references or own examples to substantiate the statement. Otherwise remove it. (It is not a question whether your own model is better, but whether there are examples models like the GPT2w model should not be used in near real-time GNSS meterology.)

Line 72. Your text says Tm it is related to several surface parameters, according to several studies. It would be good to mention at least one in addition to Ts. If you do not know another surface parameter to which Tm is strongly coupled, rephrase the sentence.

Line 74. Don't use "x" as a sign for multiplication in a formula, just write 0.72 Ts + 70.2 [K]

Line 89 Why "4 degrees   x 5degrees", not "4 degrees x 5 degrees"? (found several places in the manuscript)

Table 1. Include GPT2w in this table, to provide a comprehensive overview of the resolution of the different models, the information upon which they are based, etc.

Line 157 The full sentence doesn't read well.

Line 175. Maybe the spatial variations are "large" rather than "complicated"?

Line 190. I guess you mean better than previous "models" not "studies"?

Equation 7. Again the "x" is not necessary. Similarly the dots for multiplication inside the sines and cosines are not necessary.

Line 224. You probably mean "In addition,..", not "In contrast,.."

Figure 6. Make a test if using 2 K as minimum value of the color scale for RMSE reveals better the variations in RMSE between the different models (and regions).

Line 269. I think you mean to "find" or "identify" the best Tm model, not to verify it. You verify or validate all the Tm models, afterwards you find or identify the best, at the location of each RS.

line 304. pQ is measured in %, a drop of 0.2 doesn't sound dramatic to me.

Line 407 - 408. What are "comprehensive error sources"?  You probably mean that the GPS PWV errors are of the order 1 to 5 mm, with only part of that error being due to errors in Tm, up to 30 % at specific sites.

Notice that one writes 30 %, not 30% (common error in the manuscript).

---

## Referee Report (RR2)

Second Review Peng et al. 2018
David K. Adams
dave.k.adams@gmail.com

Recommendation:  Major Revisions

General Comments.

The English has improved substantially over the last version, but there are still many grammar errors and awkward sentences.   More importantly, the authors still need to address some of the critique of their analysis I brought up in my first review.  Firstly, the error reduction (if we can even call it that) due to their model is very small.   You need to address all of the sources of errors associated with PWV, including, very importantly, the calculation of ZHD.  Even if you cannot quantify directly these errors, you need to recognize that they exist and that they may be very large compared to your improvement in the Tm model.

These tiny error reductions of 2mm to 1mm in PWV could very easily be attributed to errors in ERA-Interim or radiosonde temperature and humidity profiles.  And because there is no standard baseline PWV value against which to judge this error reduction for a given locale, it is difficult to draw conclusions about the improvement due to the author's time-varying global gridded T s - T m model.

The authors have fastidiously avoided mentioning one of the largest sources of errors -- errors in Ps in the calculation of ZHD. An error of 1mb in the surface pressure in their equation (1)  is 2.5 times as large as an error of 1 degree C in temperature using the Bevis T s -T m equation, T m =0.72☐T s +70.2. Why is this problematic?   Let´s first assume that local barometers measure pressure perfectly, which of course isn´t true, but not much we can do about this.   Now, more importantly, the inherent uncertainties in surface pressure measurements in ERA-Interim interpolated to a local site can lead to the errors I mention above (2.5 times as large as surface temperature errors).  Of equal importance, error in antenna height relative to the height of the pressure measurement can introduce large errors. Snajdrova et al. (2005) found that 10 m of height difference approximately causes a difference of 3 mm in the ZHD.   Now imagine GPS PWV calculated with an interpolated surface pressure from ERA-Interim over very complex terrain where errors could easily be greater than 100m.

Likewise, as I insisted in my previous review.  The authors have not even mentioned the fact that ZHD doesn't take into account existing water vapor the the atmospheric column.   This error is small, but in the deep tropics the mass of water vapor can be near 4% of the total column mass.

Considering all of these sources of errors I mention about, the small reduction  (assumed)  in error due to your model is not particularly notable considering that substantial errors in surface pressure can be associated with reanalysis or radiosonde data. Finally,  the authors need to be more critical of ERA-Interim.  I have worked with these data and seen how poor they are over regions of complex terrain, for example, in the North American Monsoon region.

**Minor Comments**
Line 39  Write "GPS observations require some kinds of meteorological elements to estimate PWV ..."

Line 43  Write "are the latitude"

As I noted in my previous review, you need to clarify what ZHD is.  And the fact that it ignores water vapor in its calculation.    This is a small error (maybe more important ~4%  of total mass in equatorial regions), but you need to mention it.

Line 67.  "these models independent of real meteorological observations."   As I mentioned in my previous review, this is not typically true.  NWP models are most often initialized and constrained with some form of real-world observations.  So they are not independent in this sense.

Lin 71 Write "weather prediction."

Line 83  "without high-precision specific T s -T m equations."  Clarify what you mean here.

Line 84.  Write "Significant differences exist between oceanic  and terrestrial atmospheric properties, especially near the near the surface and within the boundary layer, in general."

Line 89-90 "However, the T s -T m relationship has time variations and can produce residuals in the static T m estimations (Yao et al., 2014a). Such residuals are not fixed in Lan's model."
It is not real clear what you are trying say here.   Lan´s model (Lan et al. 2016) is static and there for doesn´t include time variation in Ts-Tm as does Yao et al. (2014) estimation.   This is what I think you want to say.

Line 94-95  Write "...smoothing of the data,  then assess their precision,..."

Line 110  Write "are from the earth´s surface to the top of the troposphere"

Line 120 Write "...observations must be available,..."

Line 125.   As I stated in my previous review, you need to say something about the quality of ERA-Interim, particularly for regions with sparse observational data.   Humidity can be very poor in ERA-Interim and your integral depends on it.

See for example the recent work of Itterly et al. 2018

*Sensitivity of the Amazonian Convective Diurnal Cycle to Its Environment in Observations and Reanalysis*

https://doi.org/10.1029/2018JD029251

Line 129  Write " ...should be integrated through the entire atmospheric column

Line 146.  You should state here that these two are not independent even though you explore it later. Radiosondes are assimilated into ERA-Interim, so you need to state this explicitly.

Line 158 "Unreliable regression analysis results may be derived by both the T s and T m with small variations."   What are you saying here?

Line 159  "is quietly obscure".   This sounds very poetic, but I am not sure what you mean.

Line 190  " Attributed to no spatial or temporal smoothing of any data in our study, the precision and resolution of our static model, with no RMSE larger than 4.5 K, is clearly better than previous studies (Lan et al., 2016)."   Rewrite this sentence, it is not very clear.

Figure 5 is not very attractive. Neither is Figure 4.  The small numbers are a bit of a distraction.

Line 338 and throughout.  PWVs  should just be written PWV

Line 349 Some relative RMSEs were remarkably reduced. For example, at the ALIC site which is located in Australia with mean PWV of approximately 23 mm,  the relative RMSE dropped from 1.97% of PWV BTm to 1.10% of PWV VTm .

THIS IS NOT REMARKABLE!

Line 353.  "It is attributed to the wetter atmosphere in summer than in the winter."
Why would the error be larger just because the atmosphere is wetter?  You need to give a physical reason as to why this should be the case.

Line 409 "However, at some special sites, such differences could decrease by more than 30% in wetter conditions"   This is a bit misleading, your small error can decrease by 30%.

**References**

Snajdrova K., Boehm, J., Willis, P., Haas, R., and Schuh, H.:Multi-technique comparison of tropospheric zenith delays derived during the CONT02 campaign, J. Geod., 79, 613–623, doi:10.1007/s00190-005-0010-z, 200

Itterly et al. 2018
Sensitivity of the Amazonian Convective Diurnal Cycle to Its Environment in Observations and Reanalysis
https://doi.org/10.1029/2018JD029251

---

## Editor Decision (ED1)

**Editor review of "Development of time-varying global gridded Ts-Tm model for precise GPS-PWV retrieval" by Jiang et al.**

**Major remarks**

Unfortunately, the major remark addressed by David Adams has still not be addressed fully in your final revision of the manuscript: "how much is the error reduction in the GPS-PWV retrieval by using a more accurate Tm parameterization, compared to the other uncertainties in the GPS-PWV retrieval"?  Or in other words, you must state that, when using your varying Ts-Tm model,  the improvement of x% with respect to e.g. the Bevis equation will be translated in an improvement of y% in the GPS-PWV retrieval. Therefore, since you are using the Ning et al. (2016) formula (your Equation 10), you should give some typical values for the contributions of the different terms in Eq. (10) to the total uncertainty $\sigma_{PWV}$ : the first term $\sigma^2_{ZTD}$ accounts typically for around ..% of the uncertainty $\sigma_{PWV}$, the second term (with $\sigma_{Ps}$) accounts for ..% of the uncertainty $\sigma_{PWV}$, the third term (with $\sigma_C$) accounts for ..% of the uncertainty $\sigma_{PWV}$, and the fourth term ( with $\sigma_Q$) accounts for ..% of the uncertainty $\sigma_{PWV}$ (do this for different, typical values of PWV, Ps, etc.). Those different relative contributions of those terms to the total uncertainty are much more meaningful than the values shown in Table 4. By doing this, you will give the reader an idea of the importance of improving the $\sigma_Q$ for the total uncertainty $\sigma_{PWV}$ for the GPS-PWV retrieval. Up to now, this issue (raised by David Adams) has not been addressed satisfactorily by you (**and should be before the manuscript can be accepted for final publication**).  So, this aspect (and calculations) should be definitely added to your "uncertainty analysis" described in section 4.3 (lines 289-327).

I also have the feeling that this "uncertainty analysis" (lines 289-327) + the additional assessment I asked for in the previous paragraph does not fit within section 4.3 (Assessment Ts-Tm models), but better at the beginning of section 5 (GPS-PWV retrieving experiments): in section 4.3 you are comparing the differences between the different Ts-Tm models (also in Fig 9.), while in your "uncertainty analysis" you are assessing the impact of the different Ts-Tm models on the GPS-PWV retrieval. I would therefore suggest to move your "uncertainty analysis" (lines 289-327) + the additional assessment I asked for in the previous paragraph to the beginning of section 5.

I have also some remaining problems with the discussion in lines 305-312 (shown in Fig. 8). It starts with the sentence: "Then  $\sigma_{PWV}$ results were generated from the different $\sigma_Q$ estimates using equation 10." But, these important calculations to which this sentence alludes to, have never been described or shown in your manuscript, if I understand it correctly. Instead, you are focusing primarily on the uncertainty of Q itself in equation 10 (the so-called pQ, for which a formula should be provided as well), as the $T_m$ errors propagate through the determination of Q. I think that you should treat both error sources of the Tm calculation, so both the $\sigma_Q$ and the pQ, to assess the improvement of Tm varying with respect to other models (this is what an interested reader would see, instead of Fig. 8).

**Minor remarks**
- Page 1, line 17: replace to "can remove the large biases in the Bevis equation (Bevis et al., 1992) and …"

- Page 1, lines 22-23: replace "This performance is superior to the other Tm estimation models" to the percentages you find for the other models.
- Page 1, lines 23-25: rewrite after doing the assessment of how much is the error reduction in the GPS-PWV retrieval by using a more accurate Tm parameterization, compared to the other uncertainties in the GPS-PWV retrieval (see major remarks).
- Page 2, line 44: replacing "deducting" by "subtracting"
- Page 3, after line 52: give the definition of Tm here (your equation 5).
- Page 3, line 57: "It usually takes considerable amounts **of** time …"
- Page 3, line 61: "Therefore, such methods are appropriate for climate research or …" (drop "the")
- Page 3, line 69: "Many studies indicated that **the** Tm parameter has **a** relationship … "
- Page 4, lines 72-73: Change to "For example, Bevis et al. (1992) introduced the equation Tm = 0.72 Ts + 70.2 [K] after analyzing 8712 radiosonde profiles collected at 13 sites in the U.S. over two years."
- Page 4, line 75. Change to "According to Rohm et al. (2014), GPS-ZTD …"
- Page 4, line 79: "However, it is not precise enough to apply ": I do not understand what you mean here. Please rephrase.
- Page 4, lines 85-86: Change to "A global gridded Ts-Tm model has been established in Lan et al. (2016). In this model, the …"
- Page 4, lines 87-88: Change to "However, the Ts-Tm relationship is varying in time (Yao et al., 2014a), while the Lan et al. (2016) model is static."
- Page 5, line 98: "Tm is defined as a water vapour weighted mean temperature" → put this sentence and the equation (5) after line 52, page 3.
- Page 6, line 118: profiles instead of profile.
- Page 7, line 132: drop "operation".
- Page 7, line 140: Change to "However, Tm is also found not being closely related to Ts…"
- Page 7, lines 143-144: Change to "We first carried out a linear regression analysis on four years of Ts and Tm data generated from the radiosonde data and the global gridded ERA-Interim datasets, with data covering the period 2009.01 to 2012.12."
- Page 7, line 146: Change to "both analyses agree well with each other".
- Page 7-8, lines 149-151: Change to "and reach a maximum in the polar regions. The correlation coefficients drop dramatically at low latitudes. This is because Tm is stable there, showing independency of the other parameters. "
- Page 8, line 153: "… Tm varies to a lesser degree than Ts at low latitudes."
- Page 8, line 154: coefficients instead of coefficient.
- Page 8, line 154: Replace "Analyses even demonstrate …" by "We even found that …"
- Page 8, line 155: give again a reference to your figure after "Arabian Sea".
- Page 8, line 157: here, and at different other locations in the manuscript: geographical locations are denoted as 0.35°N 180°E (so the wind directions are put after the degrees, not before!!!).
- Page 8, lines 158-159: "… near the equator, because the entire variation ranges of Ts and Tm are both within 10 K. This results in a meaningless linear regression (see magenta line)."
- Page 10, line 173: change section title to "4. Development of global-gridded Ts-Tm models".

- Page 10, line 174: change to "Since the Ts-Tm relationship has large spatial variations, a global gridded Ts-Tm model is preferred for precise GPS-PWV estimations."
- Page 10, line 178: Change to "A linear formula Tm = aTs + b for the relation between Tm and Ts has been adopted in may studies."
- Page 10, line 179: please mention the time period for which you performed linear fittings of Tm versus Ts (not Ts versus Tm, I assume).
- Page 10, line 181: please add "(e.g. land, ocean)" after underlying surface.
- Page 10, lines 183-187: Change to "Constant a is smaller (approximately 0.5 to 0.7) over land in the mid to high latitudes over the Southern Hemisphere. Especially, there are abrupt changes in the values of constants a and b from land to ocean in the mid to high latitude due to different variation features of Ts and Tm (see Fig. 2). At the low latitudes, the a value is smaller than over the other regions, because of the low variations of Ts and Tm. The fitting RMSEs are within 2-4K over the mid to high latitude bands, and lower values are obtained over the oceans or at the lower latitudes. "
- Page 11, lines 189-190: Can this be rephrased to "As we did not perform any spatial or temporal smoothing of the data during the data processing, both the precision and resolution of our static model is better than other models (e.g. Lan et al., 2016)"?
- Page 12, line 197: "a precise Ts-Tm model" instead "the precise Ts-Tm model".
- Page 12, line 201: drop "of corresponding formula items". Further change "formula items" with "equations".
- Page 12, lines 203-204: Change to "Our new regression model found similar values for the coefficients a and b (of its static term) as for the static model in section 4.1, except for some differences over the oceans. In Fig. 5, besides these constants a and b, we also illustrate the amplitudes …"
- Page 13, line 209: change to "over water than over land. The estimated Tm RMSE …"
- Page 15, line 222: change to "To further assess the precision of the Ts-Tm models using …"
- Page 15, line 224: change to "independent of our model".
- Page 15, line 228: add "respectively" after "from"
- Page 15, lines 229-231: change to "When the global gridded models are employed, the radiosonde station may not be located at a grid node. Therefore, we interpolated the coefficients in the Ts-Tm equations from the …"
- Page 16, line 235: change to "$w^i$ are the interpolation coefficients, which are determined using the equation"
- Page 16, lines 238-239: change to "$\psi^i$ are computed using following formal (with latitude φ and longitude θ)"
- Page 16, line 240: please be consistent and use superscripts for the indices i, not subscripts in the equation.
- Page 16, line 241: "Considering the fact that the reanalysis grids are definite".
- Page 16, line 242: are you really referring to equation 5 here?
- Page 16, line 245: change to "Obviously, in many regions, the Bevis equation has a bad precision with the absolute bias and RSME both larger than 5 K.
- Page 16, line 245: last word: RMSEs instead of RMSE
- Page 16, line 250: at many occasions in the manuscript: bad usage of 's. Just drop them after bias and RMSE here.

- Page 16, line 254: Tm_varying 's RMSE is so awkward. Use "The RSME of Tm_varying" instead.
- Page 16, line 254: add s to second RMSE in this line.
- Page 19, in caption of Figure 7: drop 's (two times).
- Page 19, line 272: "the assumption of a normal distribution of the estimated Tm error:"
- Page 19, line 273: drop 's
- Page 20, line 283: "All the models are believed to have equivalent performances".
- Pages 20-21-22, lines 289-326: this uncertainty estimation on the GPS-PWV retrieval is out of scope here and should be moved to section 5, apart from adding different calculations (see major remarks) and rewriting.
- Page 22, line 328 and line 331: be consistent for the IGRA station names or numbers, not only here but throughout the entire manuscript. Either use the entire code EGM00062378 (like in Fig. 9) or mention just the IGRA number like IGRA station number 62378. When you give the latitude and longitude, there is no need to give them up to 4 digits after the comma. Put N and E after the number. And mention the place and country of the station as well.
- Page 23, lines 333-335: "performs as well as the Tm calculated from the radiosonde data, with small biases and capturing the variations well. The time series of Tm_GPS are smoother and cannot capture the fluctuations of the Tm time series, causing a worse accuracy than Tm_varying."
- Page 23, lines 336-337: Change "It is because" with "This can be explained by the fact that our fitting analyses are based …"
- Page 23, lines 338-339: Drop "Improvements on the reanalysis data should be performed in future".
- Page 24, line 345: "It is complicated to evaluate the GPS-PWV uncertainty here due to the lack of collocated additional…"
- Page 24, line 347: drop this sentence.
- Page 24, line 350: replace "several" by "74" and write out CDDIS (what does it stand for?).
- Page 24, line 352: add "that" after "so"
- Page 24, line 353: replace "deducted" by "subtracted" and "through" with "with".
- Page 24, line 356: add "leading to over one hundred compared points for each GPS-PWV series" after "estimates".
- Page 24, line 357: "the impact of other errors is excluded."
- Page 24, line 359: We "therefore took" instead of We "regarded". Drop "Finally, the GPS-PWV … " until "over one hundred compared points."
- Page 24, line 361: add "are" before illustrated.
- Page 24, line 362: "At most of the sites" → how many?
- Page 24, line 365: "at ALIC site (Australia), with a mean PWV of …"
- Page 25, line 369: change to "larger only in the summer season (when the PWV values are highest). Apparently, the Tm variations in summer …".
- Page 25, line 370: Drop the sentence "Furthermore, …"
- Page 25, line 371: rewrite to "some residual errors, which are removed by more than 1.0 mm in $PWV_{VTm}$.
- Page 26, line 382: "within 5 km to a nearby IGRA radiosonde station".
- Page 26, line 385: "It is worth noting that". Further on this line: drop 's after IGS site.

- Page 26, line 389: replace "occupied" with "make up the bulk of"
- Page 26, line 393-394: please be consistent in naming the radiosonde site and adapt the latitude and longitude notations (too many digits, wind direction after the numbers).
- Page 27, line 397: Some differences can reduce 1-2 mm during the wetter months. It is absolutely not clear what you mean by this statement.
- Page 27, lines 397-398: Drop the sentence "The accuracy of …", as it has not at all any added value to the analysis here.
- Page 27-28: summary and conclusion. I guess that some of the results of the extra calculations you have been asked for in my major remarks will have to be added to the summary and conclusion section.
- Page 27: you start your summary with a kind of chronological resume of your analysis. This is not how it should be done. You should first say what the major outcome of this study is (the development of your varying Tm-Ts model) and then write how can came to it (analyzing the Tm-Ts relationship based on the Ts and Tm from radiosondes and ERA-Interim).
- Page 28, line 423: "large errors": how large? Please quantify! Or at least, mention the order of magnitude.
- Page 28, lines 428-429: here again, you do not specify by how many % the errors are reduced in the GPS-PWV retrieval by using your Tm model. This is the major deficiency of your manuscript in your current form, as pointed out several times by David Adams already, and has not been adequately been taken into account. The second sentence of these lines is also too vague and should be assessed, as I've asked for.
- Page 28, line 431: drop "as well as the Ts observations".
- Page 29, line 433: drop the sentence "It could be useful for …".

---

## Author Response (AR2)

Dear associate editor and referees,

Thank you for your careful reviews. Followings are our responses to your comments.

**Responses to Associate Editor**

Dear Associate Editor,

Thank you very much for your positive comments.

We have revised our manuscript carefully. Many sentences are rewrote and many errors are revised. The missing white spaces have been added throughout the manuscript.

We address all the error sources of GPS-PWV in line 289~325. And the calculation of ZHD is discussed in particular (line 313~325). After these discussions and experiments, we point out that the GPS-PWV's error reduction due to the more precise $T_m$ estimation can be very limited. When the $\sigma_{P_s}$ is larger than 5 hPa, most of the $pQ$ values are smaller than 10 % while the error associated with the calculation of ZHD can contribute more than 80 % of the GPS-PWV's error."

**Responses to anonymous referee #3**

**General comments**

We have revised the language of our manuscript carefully. Many sentences are rewrote and many errors are revised. We actually employed the 1 by 1 degree resolution GPT2w model for the comparisons in our manuscript and we mentioned this in line 225.

**Detailed comments**

1. Remove the paragraph in lines 53 to 55, including equation 5. You do a better error budget on page 20.
*Response:* Thank you for suggestions. We delete the corresponding paragraph.

2. Lines 68 to 71 contain a statement about Tm models not including current observations not being accurate enough. You must either give references or own examples to substantiate the statement. Otherwise remove it. (It is not a question whether your own model is better, but whether there are examples models like the GPT2w model should not be used in near real-time GNSS meterology.)
*Response:* We actually studied the performance of the GPT2w model in a typhoon case. Details can be found in our previous study (Jiang et al., 2016). We add this reference to our manuscript. (Line 67)

3. Line 72. Your text says Tm it is related to several surface parameters, according to several studies. It would be good to mention at least one in addition to Ts. If you do not know another surface parameter to which Tm is strongly coupled, rephrase the sentence.

*Response:* The surface air humidity is claimed to be related to Tm by some papers (e.g. Yao et al., 2014). However the relationship is weak. We mention this in line 69~72.

4. Line 74. Don't use "x" as a sign for multiplication in a formula, just write 0.72 Ts + 70.2 [K].

*Response:* Revisions have been made throughout the manuscript.

5. Line 89 Why "4 degrees x 5degrees", not "4 degrees x 5 degrees"? (found several places in the manuscript)

*Response:* These missing white spaces have been added throughout the manuscript.

6. Table 1. Include GPT2w in this table, to provide a comprehensive overview of the resolution of the different models, the information upon which they are based, etc.

*Response:* Detail information have been added in table (1). (Line 95)

7. Line 157 The full sentence doesn't read well.

*Response:* We rewrite this sentence to "Unreliable regression analysis results may be derived if the Ts and Tm data both have small variations". (Line 155)

8. Line 175. Maybe the spatial variations are "large" rather than "complicated"?

*Response:* Revision has been made. (Line 174)

9. Line 190. I guess you mean better than previous "models" not "studies"?

*Response:* Revision has been made. (Line 189)

10. Equation 7. Again the "x" is not necessary. Similarly the dots for multiplication inside the sines and cosines are not necessary.

*Response:* Revision has been made.

$$T_m = aT_s + b + m_1 \, \boldsymbol{cos}\left(\frac{doy}{365.25}2\pi\right) + m_2 \, \boldsymbol{sin}\left(\frac{doy}{365.25}2\pi\right) + n_1 \, \boldsymbol{cos}\left(\frac{doy}{365.25}4\pi\right) +$$

$$n_2 \, \boldsymbol{sin}\left(\frac{doy}{365.25}4\pi\right) + p_1 \, \boldsymbol{cos}\left(\frac{hr}{12}\pi\right) + p_2 \, \boldsymbol{sin}\left(\frac{hr}{12}\pi\right)$$

11. Line 224. You probably mean "In addition,..", not "In contrast,.."

*Response:*   Revision has been made. (Line 223)

12. Figure 6. Make a test if using 2 K as minimum value of the color scale for RMSE reveals better the variations in RMSE between the different models (and regions).

*Response:*   Thank you for your suggestions. We replot the figures in figure 6 according to your advices. (Line 257-260)

13. Line 269. I think you mean to "find" or "identify" the best Tm model, not to verify it. You verify or validate all the Tm models, afterwards you find or identify the best, at the location of each RS.

*Response:* Revision has been made. (Line 271, 282, and 421)

14. line 304. pQ is measured in %, a drop of 0.2 doesn't sound dramatic to me.

Response: Revision has been made. "At some sites $pQ$ drops more than **20 %** from $pQ_{Bevis}$ to $pQ_{Varying.}$" (Line 310)

15. Line 407 - 408. What are "comprehensive error sources"? You probably mean that the GPS PWV errors are of the order 1 to 5 mm, with only part of that error being due to errors in Tm, up to 30 % at specific sites.

*Response:* We rewrite this sentence to "The differences between the GPS-PWV and the radiosonde PWV are approximately 1~5 mm. Some differences decrease 1~2 mm in the wetter conditions by using more precise $T_m$ models. However, the error reductions of GPS-PWV due to the $T_m$ models are very limited overall. This means that the other error sources, as we described in section 4.3, occupied the errors of GPS-PWV." (Line 424-430)

16. Notice that one writes 30 %, not 30% (common error in the manuscript).

*Response:*   We correct such errors throughout the manuscript.

**References:**

Jiang, P., Ye, S. R., Chen, D. Z., Liu, Y. Y., and Xia, P. F.: Retrieving Precipitable Water Vapor Data Using GPS Zenith Delays and Global Reanalysis Data in China, Remote Sensing, 8, 10.3390/rs8050389, 2016.

Yao, Y., Zhang, B., Xu, C., and Yan, F.: Improved one/multi-parameter models that consider seasonal and geographic variations for estimating weighted mean temperature in ground-based GPS meteorology, J. Geodesy, 88, 273-282, 10.1007/s00190-013-0684-6, 2014.

**Responses to David Adams**

**General comments**

1. The English has improved substantially over the last version, but there are still many grammar errors and awkward sentences.

*Response:* We have revised the language of our manuscript carefully. Many sentences are rewrote and many errors are revised.

2. Firstly, the error reduction (if we can even call it that) due to their model is very small. You need to address all of the sources of errors associated with PWV, including, very importantly, the calculation of ZHD. Even if you cannot quantify directly these errors, you need to recognize that they exist and that they may be very large compared to your improvement in the Tm model.

*Response:* We address all of the error sources of GPS PWV in line 289~325 and list their values in table 4 based on the summaries of other studies. We also mention the errors of ZHD calculations can be very large in line 313~325.

3. These tiny error reductions of 2mm to 1mm in PWV could very easily be attributed to errors in ERA-Interim or radiosonde temperature and humidity profiles. And because there is no standard baseline PWV value against which to judge this error reduction for a given locale, it is difficult to draw conclusions about the improvement due to the author's time-varying global gridded Ts - Tm model.

The authors have fastidiously avoided mentioning one of the largest sources of errors -- errors in Ps in the calculation of ZHD. An error of 1mb in the surface pressure in their equation (1) is 2.5 times as large as an error of 1 degree C in temperature using the Bevis Ts -Tm equation, $T_m = 0.72T_s + 70.2$. Why is this problematic? Let´s first assume that local barometers measure pressure perfectly, which of course isn´t true, but not much we can do about this. Now, more importantly, the inherent uncertainties in surface pressure measurements in ERA-Interim interpolated to a local site can lead to the errors I mention above (2.5 times as large as surface temperature errors). Of equal importance, error in antenna height relative to the height of the pressure measurement can introduce large errors. Snajdrova et al. (2005) found that 10 m of height difference approximately causes a difference of 3 mm in the ZHD. Now imagine GPS PWV calculated with an interpolated surface pressure from ERA-Interim over very complex terrain where errors could easily be greater than 100m.

Likewise, as I insisted in my previous review. The authors have not even mentioned the fact that ZHD doesn't take into account existing water vapor the atmospheric column. This error is small, but in the deep tropics the mass of water vapor can be near 4% of the total column mass.

*Response:* We discuss the errors of Ps in line 313~325. Based on Ning's summary, the uncertainty of Ps is 0.2 hPa when the surface barometer is calibrated routinely and equipped together with the GPS antenna. We enlarged it to 0.5 hPa in consideration of the possible worse performance of the surface barometers. However, we described the situations that the Ps's errors can be very large in line 318~325, and the error reductions due to our Tm models would be very limited in such cases.

4. Considering all of these sources of errors I mention about, the small reduction (assumed) in error due to your model

is not particularly notable considering that substantial errors in surface pressure can be associated with reanalysis or radiosonde data. Finally, the authors need to be more critical of ERA-Interim. I have worked with these data and seen how poor they are over regions of complex terrain, for example, in the North American Monsoon region.

*Response:* We mention the overall precision improvements of GPS-PWV introduced by the more precise Tm models can be very limited in our conclusions (Line 428~429). And we also mention the deficiencies of ERA-Interim in line 336~339.

**Minor Comments**

1. Line 39 Write "GPS observations require some kinds of meteorological elements to estimate PWV ..."

*Response:* Revision has been made. (Line 39)

2. Line 43 Write "are the latitude"

*Response:* We rewrite the sentence. (Line 43)

3. As I noted in my previous review, you need to clarify what ZHD is. And the fact that it ignores water vapor in its calculation. This is a small error (maybe more important ~4% of total mass in equatorial regions), but you need to mention it.

*Response:* We add the description about this error source in line 314~317.

4. Line 67. "these models independent of real meteorological observations." As I mentioned in my previous review, this is not typically true. NWP models are most often initialized and constrained with some form of real-world observations. So they are not independent in this sense.

*Response:* What we tried to express is that these models require no current meteorological observations which are expected to be observed together with the GPS data. To express our meaning more clearly, we rewrite the sentences to "$T_m$ at any time and any location can be estimated from these models. These models are often independent of the current meteorological observations which are required to be observed together with the GPS data."    (Line 64~66)

5. Lin 71 Write "weather prediction."

*Response:* Revision has been made. (Line 68)

6. Line 83 "without high-precision specific T s -T m equations." Clarify what you mean here.

*Response:* We rewrite the sentence to "Aside from this, some other vast areas have no specific high-precision Ts-Tm model, for example over the oceans." (Line 81-82)

7. Line 84. Write "Significant differences exist between oceanic and terrestrial atmospheric properties, especially near the near the surface and within the boundary layer, in general."

*Response:* Revision has been made. (Line 81~82)

8. Line 89-90 "However, the Ts -Tm relationship has time variations and can produce residuals in the static Tm estimations (Yao et al., 2014a). Such residuals are not fixed in Lan's model."
It is not real clear what you are trying say here. Lan´s model (Lan et al. 2016) is static and there for doesn´t include time variation in Ts-Tm as does Yao et al. (2014) estimation. This is what I think you want to say.
*Response:* We rewrite the sentences to "Actually the Ts-Tm relationship has time variation (Yao et al., 2014a). However, Lan's model is static and does not consider the time variation." (Line 82~83)

9. Line 94-95 Write "...smoothing of the data, then assess their precision,..."
*Response:* Revision has been made. (Line 89~90)

10. Line 110 Write "are from the earth´s surface to the top of the troposphere"
*Response:* Revision has been made. However, we replace your "the top of the troposphere" by our "the top level of the profile data". We think this replacement can express us more precisely. (Line 106).

11. Line 120 Write "...observations must be available,..."
*Response:* Revision has been made. (Line 116)

12. Line 125. As I stated in my previous review, you need to say something about the quality of ERA-Interim, particularly for regions with sparse observational data. Humidity can be very poor in ERAInterim and your integral depends on it.
*Response:* We add the description about the deficiencies of ERA-Interim in line 336~339.

13. Line 129 Write " ...should be integrated through the entire atmospheric column
*Response:* Revision has been made. (Line 126)

14. Line 146. You should state here that these two are not independent even though you explore it later. Radiosondes are assimilated into ERA-Interim, so you need to state this explicitly.
*Response:* We mention the relationship between the radiosondes and ERA-Interim in line 146~147.

15. Line 158 "Unreliable regression analysis results may be derived by both the Ts and Tm with small variations." What are you saying here?
*Response:* We rewrite the sentence to "Unreliable regression analysis results may be derived if the Ts and Tm data both have small variations." (Line 155)

16. Line 159 "is quietly obscure". This sounds very poetic, but I am not sure what you mean.
*Response:* We rewrite the sentences to "As the blue dots show, the $T_s$-$T_m$ relationship is weak in the areas near the equator.

It is because that the entire variation ranges of $T_s$ and $T_m$ are both below 10 K" (Line 157~158).

17. Line 190 " Attributed to no spatial or temporal smoothing of any data in our study, the precision and resolution of our static model, with no RMSE larger than 4.5 K, is clearly better than previous studies (Lan et al., 2016)." Rewrite this sentence, it is not very clear.

*Response:* We rewrite the sentences to "Meanwhile, there is no RMSE larger than 4.5 K in the results of our model. The precision and the resolution of our static model is clearly better than previous models (Lan et al., 2016). It is because that we performed no spatial or temporal smoothing of the data during the data processing." (Line 188~190)

18. Figure 5 is not very attractive. Neither is Figure 4. The small numbers are a bit of a distraction.

*Response:* We delete the numbers in figure 4 and 5.

19. Line 338 and throughout. PWVs should just be written PWV

*Response:* Revisions have been made throughout the manuscript.

20. Line 349 Some relative RMSEs were remarkably reduced. For example, at the ALIC site which is located in Australia with mean PWV of approximately 23 mm, the relative RMSE dropped from 1.97% of PWV BTm to 1.10% of PWV VTm . THIS IS NOT REMARKABLE!

*Response:* We delete the sentence "Some relative RMSEs were remarkably reduced", and we add a sentence "We found that some relative RMSE could reduce more than 2 % from $PWV_{BTm}$ to $PWV_{VTm}$" in line 367.

21. Line 353. "It is attributed to the wetter atmosphere in summer than in the winter." Why would the error be larger just because the atmosphere is wetter? You need to give a physical reason as to why this should be the case.

*Response:* We rewrite the sentences to "It is because that the $T_m$'s variations in summer are not modeled well by both Bevis model and the latitude-related model. Furthermore, the higher PWV values in summer enlarge the PWV differences." (Line 369~370)

22.    Line 409 "However, at some special sites, such differences could decrease by more than 30% in wetter conditions" This is a bit misleading, your small error can decrease by 30%.

*Response:* We rewrite the sentences to "The differences between the GPS-PWV and the radiosonde PWV are approximately 1~5 mm. Some differences decrease 1~2 mm in the wetter conditions by using more precise $T_m$ models" to avoid the possible misleading" in line 427-428. Corresponding revision has been made in the abstract in line 25.

---

## Author Response (AR3)

Dear editor Roeland Van Malderen,

Thank you very much for your careful review and good suggestions to improve our manuscript. Our responses to your comments are as follows.

**Response to major remarks**

We add many calculations to the analyses of GPS-PWV uncertainty (lines 307~384), and these analyses are rewrote in section 5.1 instead of section 4.4. Both the $\sigma_Q$ and $p_Q$ are assessed carefully with different typical values. The analysis of $\sigma_Q$ is given in lines 314~328, and the contribution of the uncertainty associated with $Q$ to the total GPS-PWV uncertainty is investigated as well as the contributions of other terms (e.g. $p_{ZTD}$, $p_{Ps}$, and $p_C$) in lines 329~374 with the typical values listed in table 4. New formulas of $p_Q$, $p_{ZTD}$, $p_{Ps}$, and $p_C$ are added in equation (12). The quantitative results are given in detail to avoid the vague descriptions. The conclusions drawn from these new added calculations are added to section 6 "Summery and conclusion" (lines 454-468) and the abstract (lines 20~30).

**Response to minor remarks**

The language revisions have been made throughout the manuscript. The changes are shown in the endings of this file. Aside from the language revisions, responses to the major comments are as follows.

1. **Comment:** Page 1, lines 22-23: replace "This performance is superior to the other Tm estimation models" to the percentages you find for the other models

   **Response:** These sentences are rephrased in lines 25~29.

2. **Comment:** Page 1, lines 23-25: rewrite after doing the assessment of how much is the error reduction in the GPS-PWV retrieval by using a more accurate Tm parameterization, compared to the other uncertainties in the GPS-PWV retrieval (see major remarks).

   **Response:** The conclusions drawn from the new added calculations are added to the abstract (lines 20~30).

3. **Comment:** Page 3, after line 52: give the definition of Tm here (your equation 5).

   **Response:** The definition of Tm is moved to section 1 (lines 57~63).

4. **Comment:** Page 16, line 250: at many occasions in the manuscript: bad usage of 's.

   **Response:** Revisions have been made throughout the manuscript.

5. **Comment:** Page 22, line 328 and line 331: be consistent for the IGRA station names or numbers, not only here but throughout the entire manuscript. Either use the entire code EGM00062378 (like in Fig. 9) or mention just the IGRA number like IGRA station number 62378. When you give the latitude and longitude, there is no need to give them up to 4 digits after the comma. Put N and E after the number. And mention the place and country of the station as well.

   **Response:** Revisions have been made throughout the manuscript. A radiosonde station is described with its IGRA number, location and area, e.g. the IGRA station No.62378 (29.86 °N 31.34 °E, in Egypt). And a GPS station is mentioned with its name, location and area, e.g. the ALIC site (23.67 °S 133.89 °E, in Australia).

6. **Comment:** Page 27, line 397: Some differences can reduce 1-2 mm during the wetter months. It is absolutely not clear what you mean by this statement.

   **Response:** Sentences are rephrased in lines 433~435.

7. **Comment:** Page 27-28: summary and conclusion. I guess that some of the results of the extra calculations you have been asked for in my major remarks will have to be added to the summary and conclusion section.

   **Response:** The conclusions drawn from the new added calculations are added in lines 454-468.

8. **Comment:** Page 27: you start your summary with a kind of chronological resume of your analysis. This is not how it should be done. You should first say what the major outcome of this study is (the development of your varying Tm-Ts model) and then write how can came to it (analyzing the Tm-Ts relationship based on the Ts and Tm from radiosondes and ERA-Interim).

   **Response:** We rewrite the summary and conclusion in lines 442~471.

9. **Comment:** Page 28, line 423: "large errors": how large? Please quantify! Or at least, mention the order of magnitude.

   And Page 28, lines 428-429: here again, you do not specify by how many % the errors are reduced in the GPS-PWV retrieval by using your Tm model. This is the major deficiency of your manuscript in your current form, as pointed out several times by David Adams already, and has not been adequately been taken into account. The second sentence of these lines is also too vague and should be assessed, as I've asked for.

   **Response:** We give the quantitative results throughout the revised manuscript to avoid the vague descriptions.

[revised manuscript text omitted]

批注 [jiang18]: Sentence changed

批注 [jiang19]: Sentence changed

批注 [jiang20]: Sentence revised

批注 [jiang21]: Sentence revised

批注 [jiang22]: Sentence changed

批注 [jiang23]: Sentence changed

批注 [jiang24]: Sentence changed

批注 [jiang25]: Reference to figure 1 has been added

批注 [jiang26]: Revisions have been made throughout the manuscript

批注 [jiang27]: Sentence changed

[Figure]

**Figure 1: Correlation coefficients between $T_s$ and $T_m$ generated from radiosonde data (dots) and ERA-Interim reanalysis datasets (color-filled contours) over a period of 4 years from 2009 to 2012.**

[Figure]

**Figure 2: Denary logarithm of the standard deviation of (top) $T_s$ and (bottom) $T_m$ generated from the ERA-Interim data covering the year 2009 to 2012. Temperature unit is Kelvin.**

[Figure]

**Figure 3:** $T_s$-$T_m$ scatter plots at two locations: (blue dots) 0.35° N 180.00° E and (red dots) 70.53° N 180.00° E, the magenta and green lines are their linear fitting curves. Temperature unit is Kelvin.

**4. Development of global-gridded $T_s$-$T_m$ models**

Since the $T_s$-$T_m$ relationship has large spatial variations, a global gridded $T_s$-$T_m$ model is preferred for precise GPS-PWV estimations. In this section, a static global gridded model and a time-varying global gridded model are established and assessed.

批注 [jiang28]: Sentence changed

**4.1 Static global-gridded $T_s$-$T_m$ model**

A linear formula $T_m = aT_s + b$ for the relation between $T_m$ and $T_s$ has been adopted in many studies. Based on the $T_s$ and $T_m$ products from the ERA-Interim data covering the year 2009 to 2012, we performed linear fittings of $T_m$ versus $T_s$ on each grid point. Then, the slope constant ($a$), the intercept constant ($b$) and the fitting root mean square error (RMSE) of each linear expression were calculated and contoured in figure 4. The $a$ and $b$ values are related to the latitude as well as the underlying surface (e.g. land, ocean). In the mid-high latitudes over the Northern Hemisphere, constant $a$ value varies from 0.6 to 0.8, and constant $b$ is approximately 100~50 over most of the continents. The constants in the Bevis equation are within these value ranges. Constant $a$ is smaller (approximately 0.5~0.7) over land at the mid to high latitudes over the Southern Hemisphere. Especially, there are abrupt changes in the values of constants $a$ and $b$ from land to ocean at the mid to high latitudes due to the different variation features of $T_s$ and $T_m$ (see figure 2). At the low latitudes, the $a$ value is smaller than over the other regions, because of the low variations of $T_s$ and $T_m$. The fitting RMSEs are within 2~4 K over the mid to high latitude lands, and lower values are obtained over the oceans or at the low latitudes. The reason for the low RMSE around the equator is the smaller fluctuation of $T_m$. Meanwhile, there is no RMSE larger than 4.5 K in the results of our model. As we did not perform any spatial or temporal smoothing of the data during the data processing, both the precision and resolution of our static model is better than other models (e.g. Lan et al., 2016).

批注 [jiang29]: The time period for the linear fittings of Tm versus Ts has been mentioned

批注 [jiang30]: Explanation added

批注 [jiang31]: Sentences rephrased

批注 [jiang32]: Sentence revised

[revised manuscript text omitted]

批注 [jiang51]: IGRA station numbers have been adopted throughout the entire manuscript, e.g. the IGRA station No.62378. The description about the position of a station has been revised throughout the manuscript.

批注 [jiang52]: Sentence rephrased.

批注 [jiang53]: Sentence rephrased.

批注 [jiang54]: The sentence "Improvements on the reanalysis data should be performed in future" has been deleted

**Figure 8:** *$T_m$ series of $T_{m\_Bevis}$, $T_{m\_LatR}$, $T_{m\_static}$, $T_{m\_varying}$, $T_{m\_GPT2w}$ and $T_{m\_RS}$ at the IGRA station (top) No.62378 and (bottom) No.40841. Temperature unit is Kelvin.*

**5. GPS-PWV retrieving experiments**

GPS-PWV has different error sources with different properties. It is complicated to evaluate the GPS-PWV uncertainty here due to the lack of collaborated additional independent techniques to monitor water vapor at the GPS site.

**5.1 Theoretical analysis of the GPS-PWV uncertainty**

A comprehensive research on the uncertainty of GPS-PWV has been carried out by Ning et al. (2016). The uncertainties of the ZTD, ZHD and conversion factor $Q$ have been studied in detail. The total uncertainty of GPS-PWV is:

$$\sigma_{PWV} = \frac{1}{Q}\sqrt{\sigma_{ZTD}^2 + \left(\frac{2.2767\sigma_{Ps}}{f(\varphi,H)}\right)^2 + \left(\frac{P_s\sigma_c}{f(\varphi,H)}\right)^2 + \left(PWV \cdot \sigma_Q\right)^2} \tag{10}$$

where $\sigma_{PWV}$, $\sigma_{ZTD}$, $\sigma_{Ps}$, and $\sigma_Q$ are respectively the uncertainties of GPS-PWV, ZTD estimation, $P_s$ observations and conversion factor $Q$. $\sigma_c = 0.0015$ denotes the uncertainty of constant $C = 2.2767$ in equation (1), $PWV$ is the value of GPS-PWV, and

$$\sigma_Q = 10^{-6}\rho_w R_v \sqrt{\left(\frac{\sigma_{k_3}}{T_m}\right)^2 + \sigma_{k_2'}^2 + \left(k_3\frac{\sigma_{T_m}}{T_m^2}\right)^2} \tag{11}$$

where $\sigma_{k_3} = 0.012 \times 10^5$ K$^2$ hPa$^{-1}$, $\sigma_{k_2'} = 2.2$ K hPa$^{-1}$, and $\sigma_{T_m}$ denote respectively the uncertainties of $k_3$, $k_2'$ and $T_m$ in equation (4). The variation of $\sigma_Q$ with the value of $T_m$ and $\sigma_{T_m}$ is depicted in figure 9. Assuming the $T_m$ is 280 K, we find that the $\sigma_Q$ increases by over 60 % (from 0.069 to 0.112) as the $\sigma_{T_m}$ raises from 3.0 K to 5.0 K. However, the $\sigma_Q$ is less sensitive to the value of $T_m$. The $\sigma_Q$ raises only by 17.96 % (about from 0.061 to 0.075) as the value of $T_m$ drops from 300 K to 270 K with $\sigma_{T_m} = 3.0$ K.

Ning et al. (2016) assumed the $T_m$ were obtained from NWP models so the uncertainty of $T_m$ was set to be small ($\sigma_{T_m} = 1.1$ K). However, as shown in section 4.3, the uncertainties of $T_m$ from different $T_m$ models are significantly larger at the radiosonde stations. For each radiosonde station, we calculated the mean value of $T_m$ and assigned the $\sigma_{T_m}$ with the RMSEs of $T_m$ given in figure 6. Then we obtained the $\sigma_Q$ in equation (11). Our statistics indicate that the $\sigma_Q$ using our

批注 [jiang55]: Sentence revised

批注 [jiang56]: The uncertainty analysis has been moved here

varying $T_s$-$T_m$ model decreases by average 19.26 %, 17.77 %, 7.79 % and 18.67 % with respect to the $\sigma_Q$ respectively using the $T_{m\_Bevis}$, $T_{m\_LatR}$, $T_{m\_static}$, and $T_{m\_GPT2w}$. For example, at the IGRA station No.42724 (22.88° N 91.25° E, in India), $\sigma_Q$ drops by 53 % from 0.141 of the $T_{m\_Bevis}$ to 0.066 of the $T_{m\_varying}$.

[Figure]

**Figure 9. Variation of the uncertainty of Q with the value of $T_m$ and the uncertainty of $T_m$**

The uncertainty of $Q$ will be propagated to the total uncertainty of GPS-PWV according to equation (10). We obtained the contributions of the different terms in equation (10) to the total GPS-PWV uncertainty. The contribution of one term is measured by the percentage it accounts for the total $\sigma_{PWV}$. The percentages are computed using formulas:

$$p_{ZTD} = \frac{(\sigma_{ZTD}/Q)^2}{\sigma_{PWV}^2}, \quad p_{Ps} = \frac{\left[2.2767\sigma_{Ps}/\left(f(\varphi,H)Q\right)\right]^2}{\sigma_{PWV}^2}, \quad p_C = \frac{\left[P_s\sigma_c/\left(f(\varphi,H)Q\right)\right]^2}{\sigma_{PWV}^2}, \quad p_Q = \frac{\left(PWV\cdot\sigma_Q/Q\right)^2}{\sigma_{PWV}^2} \tag{12}$$

[revised manuscript text omitted]